# Driving forces behind phase separation of the carboxy-terminal domain of RNA polymerase II

David Flores-Solis [1], Irina P. Lushpinskaia[1], Anton A. Polyansky[2,3], Arya Changiarath[4,5], Marc Boehning[6], Milana Mirkovic[2,3], James Walshe[6], Lisa M. Pietrek[7], Patrick Cramer[6], Lukas S. Stelzl [4,5,8], Bojan Zagrovic [2,3] & Markus Zweckstetter [1,9] ✉

Eukaryotic gene regulation and pre-mRNA transcription depend on the carboxy-terminal domain (CTD) of RNA polymerase (Pol) II. Due to its highly repetitive, intrinsically disordered sequence, the CTD enables clustering and phase separation of Pol II. The molecular interactions that drive CTD phase separation and Pol II clustering are unclear. Here, we show that multivalent interactions involving tyrosine impart temperature- and concentration-dependent self-coacervation of the CTD. NMR spectroscopy, molecular ensemble calculations and all-atom molecular dynamics simulations demonstrate the presence of diverse tyrosine-engaging interactions, including tyrosine-proline contacts, in condensed states of human CTD and other low-complexity proteins. We further show that the network of multivalent interactions involving tyrosine is responsible for the co-recruitment of the human Mediator complex and CTD during phase separation. Our work advances the understanding of the driving forces of CTD phase separation and thus provides the basis to better understand CTD-mediated Pol II clustering in eukaryotic gene transcription.

Cellular organization processes depend on the formation of membrane-less organelles/biomolecular condensates[1–3]. Increasing evidence suggests an important role of the phase separation of proteins and nucleic acids in gene transcription[4–8]. In agreement with RNA polymerase II (Pol II)-associated condensation, transcriptionally active clusters of Pol II in the nucleus of eukaryotic cells, the so-called "transcription hubs", exhibit transient, highly dynamic nature[4,9–12]. An important role in the formation of Pol II clusters is played by the intrinsically disordered carboxy-terminal domain (CTD) of the largest subunit of Pol II, RPB1. The CTD is critical for pre-mRNA synthesis and co-transcriptional processing[13] and can undergo liquid-liquid phase separation in vitro[14]. The mechanistic basis of CTD phase separation, Pol II clustering, and thus the formation of eukaryotic transcription hubs is however largely unknown.

The CTD is a low-complexity sequence conserved among organisms and comprising the consensus heptad repeat sequence

[1]German Center for Neurodegenerative Diseases (DZNE), Von-Siebold Straße 3A, 35075 Göttingen, Germany. [2]Max Perutz Labs, Vienna Biocenter Campus (VBC), Campus Vienna Biocenter 5, 1030 Vienna, Austria. [3]University of Vienna, Center for Molecular Biology, Department of Structural and Computational Biology, Campus Vienna Biocenter 5, 1030 Vienna, Austria. [4]Faculty of Biology, Johannes Gutenberg University Mainz (JGU), Gresemundweg 2, 55128 Mainz, Germany. [5]KOMET1, Institute of Physics, Johannes Gutenberg University Mainz (JGU), Staudingerweg 9, 55099 Mainz, Germany. [6]Department of Molecular Biology, Max Planck Institute for Multidisciplinary Sciences, Am Faßberg 11, 37077 Göttingen, Germany. [7]Department of Theoretical Biophysics, Max Planck Institute of Biophysics, Max-von-Laue Straße 3, 60438 Frankfurt am Main, Germany. [8]Institute of Molecular Biology (IMB), 55128 Mainz, Germany. [9]Department of NMR-based Structural Biology, Max Planck Institute for Multidisciplinary Sciences, Am Faßberg 11, 37077 Göttingen, Germany. ✉e-mail: Markus.Zweckstetter@dzne.de

$Y_1S_2P_3T_4S_5P_6S_7$[15,16]. Human CTD (hCTD) contains 52 heptad repeats with a divergent distal region (Fig. 1a). The yeast *Saccharomyces cerevisiae* CTD (yCTD) resembles the first 26 repeats of the human protein. Human and yeast CTD sequences undergo concentration-dependent phase separation in the presence of crowding agents[14]. The formed droplets are liquid-like and incorporate intact Pol II[14]. In addition, the propensity of Pol II for clustering in the nucleus depends on the number of heptad repeats: truncation of hCTD to the length of yCTD decreases detectable Pol II hubs and chromatin association in human cells, while repeat extension increases Pol II clustering. Consistent with CTD-dependent Pol II clustering, CTD length modulates transcriptional bursting[17]. CTD interactions are thus important for the formation of Pol II clusters at active genes[14].

Many transcription factors and enzymes generate a complex yet specific pattern of regulation of Pol II at different stages of gene transcription[13]. In vitro experiments showed that CTD droplets are dissolved through CTD phosphorylation by the transcription initiation factor IIH kinase CDK7[14]. CDK7 preferentially phosphorylates $S_5$ and $S_7$ in the heptad repeat. Hypo- and hyperphosphorylated hCTD also phase separates when combined with the human Mediator complex (hMED) during the initiation and elongation steps of transcription[10,12,18]. In addition, the degradation of Mediator in cells causes the disassembly of large clusters of hypophosphorylated

Pol II[19], suggesting orchestrated processes of Pol II/Mediator condensation regulated by phosphorylation[5,18].

Using a combination of phase separation assays, NMR spectroscopy, molecular ensemble calculations, all-atom molecular dynamics simulations, and site-directed mutagenesis, we provide insight into the mechanistic basis of CTD phase separation. We show that a broad spectrum of interactions, including tyrosine-proline interactions, drives CTD phase separation and are abundant in the condensed states of other low-complexity proteins. We further show that the human CTD phase separates together with the 1.37 MDa human Mediator complex.

## Results

### Self-coacervation of CTD

To gain insight into the nature of multivalent interactions that drive CTD phase separation, we expressed and purified recombinant constructs of hCTD (382 residues) and yCTD (196 residues). To reach high purity, the proteins were purified by reversed-phase HPLC in the last purification step. The obtained proteins were free of tags (Supplementary Fig. 1).

hCTD and yCTD proteins were subjected to phase separation experiments. To establish the temperature- and ionic-strength-dependent phase diagram, we used dynamic light scattering (DLS)

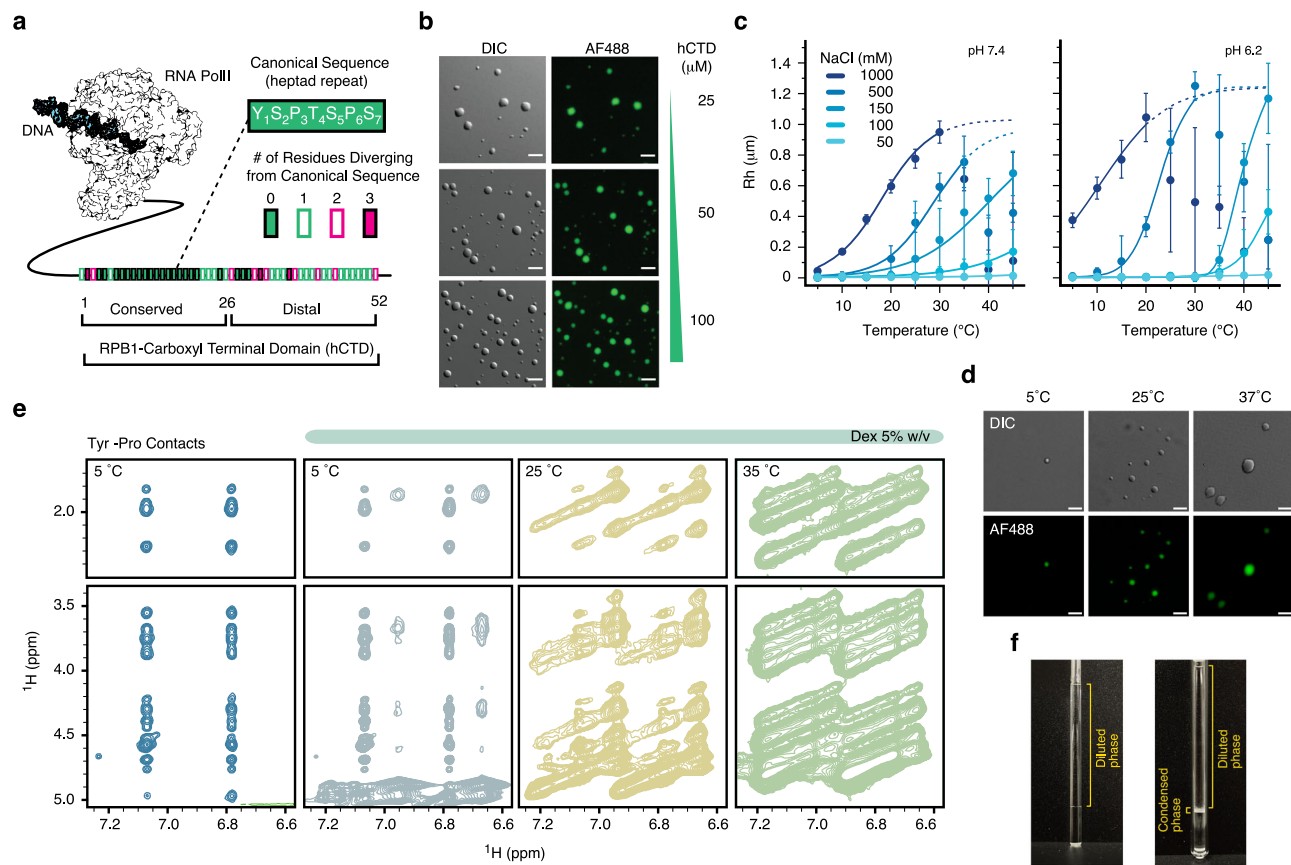

**Fig. 1 | Phase separation of human CTD. a** Schematic representation of human RNA Pol II illustrating the conserved heptad repeats YSPTSPS of the CTD of RPB1, the largest subunit of Pol II. Variations from the YSPTSPS repeat sequence are color coded. **b** Micrographs demonstrating concentration-sensitive phase separation of hCTD in the absence of molecular crowding agents (300 mM NaCl, 25 mM HEPES, 1.0 mM TCEP, pH 7.4). **c** Influence of temperature, pH, and ionic strength to induce hCTD phase separation (in 25 mM HEPES, 1.0 mM TCEP) monitored by dynamic light scattering. No crowding agents were used. Solid lines represent sigmoidal regression; dotted lines indicate tendencies in conditions where sedimentation occurred. Dots represent mean values (*n* = 3) and error bars represent ± std for

independent measurements. **d** Droplet morphologies of hCTD (25 μM) in high salt conditions. The pictures correspond to 1 M NaCl in 25 mM HEPES, 1.0 mM TCEP (pH 7.4) at three different temperatures. **e, f** Interresidue contacts in two-dimensional $^1$H-$^1$H NOESY spectra of yCTD in isotropic mixed conditions (5 °C without dextran; blue) and in conditions of phase separation (5 % dextran and increasing temperature). Cross-peaks between aromatic Tyr protons and aliphatic side-chain protons (vertical scale) are displayed in (**e**). The NMR tubes in (**f**) show the formation of a macroscopic condensate prior to recording the NOESY spectra in the presence of dextran and increased temperature. Micrographs are representative of 3 independent biological replicates. Scale bar, 5 μm.

and microscopy (Fig. 1b, c). In the absence of crowding agents, we consistently observed the formation of CTD-enriched droplets at minimal concentrations of ~25 and ~100 μM for hCTD and yCTD, respectively (Fig. 1b, c and Supplementary Fig. 2a, b). yCTD mostly remained monomeric at ~25 μM with some signs of potential oligomerization (Supplementary Fig. 2a). The ~4-fold lower critical concentration of phase separation of hCTD when compared to yCTD supports the importance of multivalent interactions between the heptad repeats for CTD phase separation.

Temperature-dependent DLS experiments further showed that phase separation of hCTD, as well as yCTD, display lower critical solution temperature (LCST) behavior. At 5 °C, pH 7.4, the CTD solutions were uniformly mixed (Fig. 1c and Supplementary Fig. 2b). Upon temperature increase, phase separation occurred and droplets formed as confirmed by microscopy (Fig. 1b–d). The experiments further showed that the critical temperature for CTD phase separation depends on ionic strength. Hypertonic solutions (500 mM and 1 M NaCl) decreased the critical temperature for phase separation, while low ionic strength increased it to the degree that we did not detect phase separation of 25 μM hCTD in the presence of 50 mM NaCl, pH 7.4 (Fig. 1c, top). The LCST behavior of CTD is in agreement with a low content of charged amino acids[20].

We then lowered the pH from 7.4 to 6.2, closer to the theoretical pI (5.8) of yCTD (Supplementary Fig. 2c), and redefined the phase diagram (Fig. 1b, c and Supplementary Fig. 2a, b, d). CTD again displays LCST behavior, in which phase separation occurs above a critical temperature. When compared to pH 7.4, the critical temperature is however shifted to lower values (Fig. 1c).

We also note that at high ionic strength and increased temperature CTD droplets rapidly sediment, complicating DLS analysis. Rapid sedimentation and adherence of the CTD droplets to surfaces can be decreased by adding dextran to the solution. 5% w/v of the molecular crowding agent dextran also promotes CTD phase separation (Supplementary Fig. 2b). The combined data show that hCTD as well as yCTD undergo temperature- and concentration-dependent self-coacervation.

## Multivalent tyrosine interactions in CTD condensates

To gain insight into molecular interactions inside CTD condensates, we used NMR spectroscopy (Fig. 1e). We prepared a concentrated solution of yCTD (2.1 mM yCTD, 300 mM NaCl), added 5 % w/v dextran to further promote phase separation, incubated the sample at 5, 25, and 35 °C to control phase separation, and used centrifugation to obtain a macroscopic condensate at the bottom of the NMR tube (Fig. 1f). In addition, we prepared a second sample (2.1 mM yCTD, 300 mM NaCl) that lacked dextran and was kept at 5 °C, i.e. low temperature which impairs CTD phase separation (Fig. 1f). For each condition, we recorded two-dimensional $^1$H-$^1$H-NOESY experiments (Fig. 1e).

In the uniformly mixed phase at 5 °C without dextran, we observed through-space correlations between the aromatic protons of tyrosine (Tyr) residues (horizontal scale in Fig. 1e) and the side-chain protons of prolines (Pro), serines (Ser) and threonines (Thr) (vertical scale in Fig. 1e). The interactions are predominantly intramolecular because oligomerization and/or phase separation were not detected in the uniformly mixed sample at 5 °C.

Next, we compared the $^1$H-$^1$H-NOESY spectrum of the uniformly mixed sample to spectra recorded for yCTD in the presence of dextran. At 5 °C, the pattern of tyrosine-mediated contacts with identical chemical shifts were present (Fig. 1e). In addition, we observed cross-peak patterns, which were up-field shifted by ~0.25 ppm in both $^1$H dimensions. At higher temperatures, additional Tyr-ring spin systems appeared, which were shifted either down-field or up-field when compared to the cross-peak pattern in the uniformly mixed sample. At 35 °C, separated cross-peaks were replaced by streaks of interresidue correlations (Fig. 1e and Supplementary Fig. 2e, g).

The observed heterogeneity in chemical shifts might be due to a combination of inhomogeneities inside the condensate and associated exchange processes plus different magnetic susceptibilities from emerging droplets in the sample. In addition, only a small macroscopic condensate was visible at the bottom of the NMR tube (despite the high yCTD concentration and large sample volume; Fig. 1f), further contributing inhomogeneities at the interface between the condensate and the dilute phase. The presence of distinct chemical environments in the phase separated sample was confirmed by two-dimensional $^1$H-$^{13}$C-HSQC correlations of the aromatic rings of Tyr (Supplementary Fig. 2g). Despite the heterogeneity in chemical environment, the NMR analysis demonstrates that Tyr-Pro, Tyr-Ser and Tyr-Thr contacts are abundant inside CTD condensates. The contacts can be either intra- or intermolecular.

## Dynamic structure of CTD heptad repeats

To further understand the nature of the multivalent CTD interactions, we studied the structural biases in the CTD using complementary structure-sensitive NMR probes. First, we recorded two-dimensional $^1$H-$^{15}$N-TROSY spectra for $^{15}$N-labeled hCTD and yCTD in the dilute, non-phase separated state (Fig. 2a). The spectra display low $^1$H chemical shift dispersion indicative of the lack of α-helix/β-strands and/or tertiary structure. Superposition of the spectra of hCTD and yCTD shows that five cross-peaks have very high intensity and have identical chemical shifts in hCTD and yCTD (Fig. 2a inset). Additional weaker peaks likely arise from residues in non-conserved heptad repeats and from cis-trans isomerization of prolines. The observation of five strong cross-peaks suggests that the five non-proline residues of conserved heptad repeats ($Y_1$, $S_2$, $T_4$, $S_5$ & $S_7$) experience identical chemical environments.

To support this interpretation, we prepared shorter CTD fragments comprising one to six conserved heptad repeats (named 1R- to 6R-CTD). For 1R-CTD, we detected five cross-peaks, and for 2R-CTD, ten cross-peaks in agreement with the number of non-proline residues. For 3R-CTD, the spectrum appeared very similar to 2R-CTD, but displayed additional slightly shifted signals (Supplementary Fig. 3). Inclusion of additional repeats only increased the intensity of the cross-peaks, which were also most intense in the spectra of hCTD/yCTD. We then determined the sequence-specific assignment of the cross-peaks in 1R-, 2R- and 3R-CTD using two-dimensional $^1$H-$^1$H TOCSY and NOESY spectra. By gradually increasing CTD length, we were able to assign the residues of 1R-, 2R- and 3R-CTD (Fig. 2b and Supplementary Fig. 3). The assignment confirmed that the most intense NMR signals in hCTD and yCTD arise from the conserved heptad repeats.

The identical chemical shifts of the second repeat of 3R-CTD and the conserved heptad repeats of hCTD and yCTD suggest that the structural properties of conserved heptad repeats are similar, i.e., a well-defined structural motif is repeated in the conformational ensemble of hCTD/yCTD. Consistent with this hypothesis, a comparison of the two-dimensional NOESY spectra of yCTD and 3R-CTD revealed similar cross-peak patterns between the aromatic ring protons of Tyr and the side-chain protons of Pro, Ser and Thr (Fig. 2c). Some of the strongest signals were seen between the Hε ring protons of Tyr and the Hγ protons of Pro (Fig. 2c, Tyr-HE & Pro-HG peaks ~1.33 times the average NMR signal in the corresponding stripe between 0 to 5 ppm). Rotating-frame exchange spectroscopy experiments confirmed that the cross-peaks rise from direct contacts and not from exchange (Supplementary Fig. 4a). Tyr-Pro, Tyr-Ser and Tyr-Thr contacts were also observed in yCTD at both 5 °C and 37 °C (Supplementary Fig. 4b).

We next measured residual $^1$H-$^{15}$N dipolar couplings (D(Hz)), which report on the structure and dynamics of the protein backbone (Supplementary Fig. 4c). In the case of an IDP populating predominantly extended structure, the N-H vectors are predominantly orthogonal with respect to the main chain and thus will display the same sign. Consistent with a mainly extended structure, the residual dipolar couplings in 2R- and 3R-CTD have the same sign (Fig. 2d).

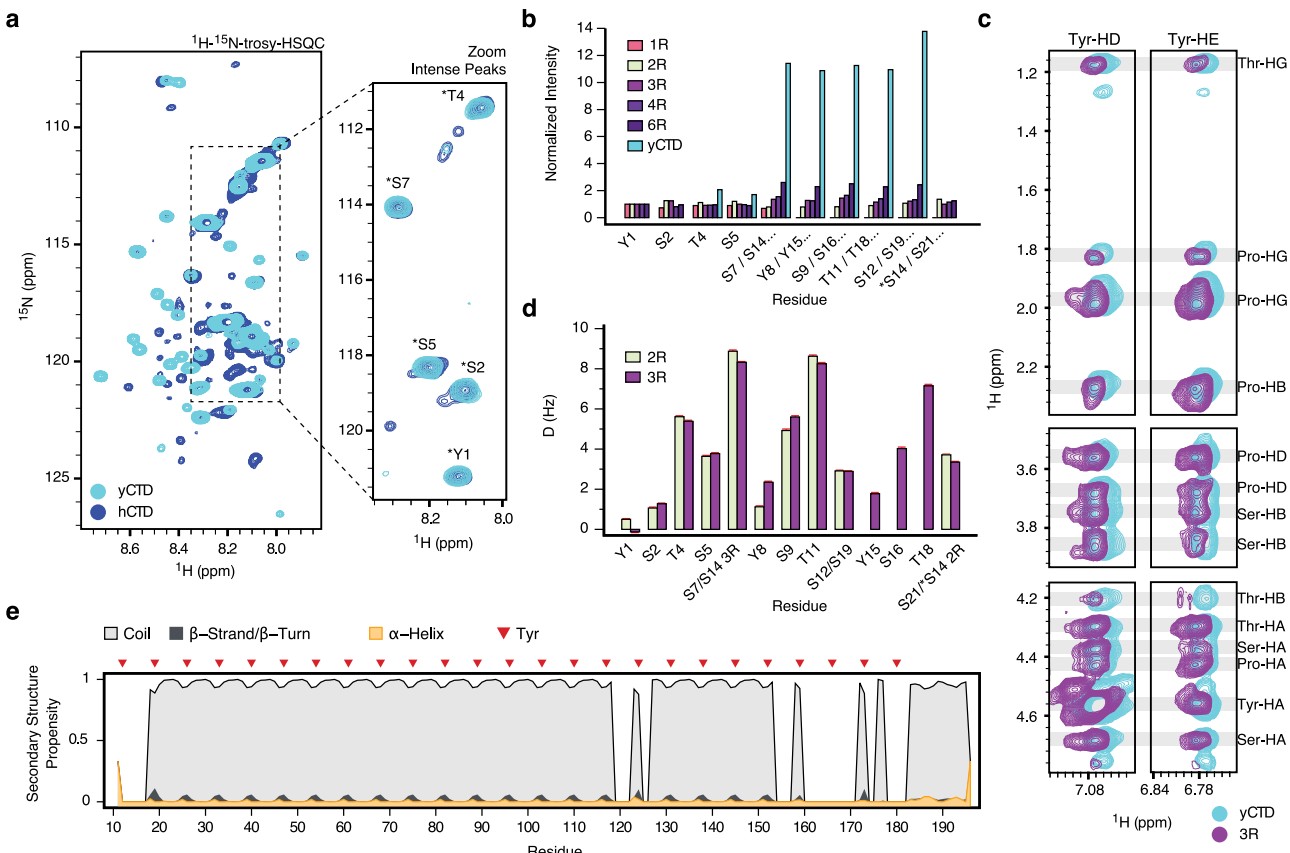

**Fig. 2 | Tyrosine-proline contacts in CTD heptad repeats. a** Superposition of $^1$H-$^{15}$N- TROSY spectra of human (dark blue) and yeast (cyan) CTD. The region highlighted displays the most intense peak resonances from residues of the canonical heptad repeats (above 50% of the initial threshold). Typical chemical shifts from canonical residues (*) are indicated in the inset. **b** Residue-specific normalized cross peak intensities observed in $^1$H-$^{15}$N-HSQC spectra of yCTD and CTD peptides composed of one to six canonical heptad repeats YSPTSPS. For longer sequences, the cross peaks of individual heptad repeats overlap (labeled e.g. as "S7/S14…"). **c** NOE contacts between Tyr ring protons and aliphatic proline protons in two-dimensional $^1$H-$^1$H NOESY spectra of 3R-CTD (purple) and yCTD (cyan) at non-phase separating conditions (5 °C). In addition to Tyr-Pro contacts, Tyr-Thr and Tyr-Ser cross-peaks were assigned in the NOESY spectrum of 3R-CTD. **d** $^1$H-$^{15}$N residual dipolar couplings (RDCs) for 2R- and 3R-CTD. Due to the increased resolution in the IPAP-HSQC experiments, RDCs could be determined for individual residues in the repeats. **e** Secondary structure propensity in yCTD derived from experimental NMR chemical shifts. The location of Tyr residues is marked by red triangles. Non-assigned/overlapping residues were excluded from the analysis.

In addition, the heptad residues display a specific pattern of D values rising from the heptad position $Y_1$ to $S_2$ and $T_4$ followed by a decrease at position $S_5$. The variation in the magnitude of the residual dipolar coupling arises from either differences in dynamics (higher dynamics resulting in smaller residual dipolar couplings) or specific conformational properties of the CTD ensemble. For example, the small residual dipolar coupling values of the Tyr residues could arise from a turn conformation. Importantly, a comparison of 2R- and 3R-CTD reveals similar dipolar coupling patterns for the residues in the heptad repeats 1 and 2 in both peptides.

We also recorded triple-resonance experiments for $^{13}$C/$^{15}$N-labeled yCTD. Careful manual analysis of the spectra allowed us to determine the backbone resonance assignment of many residues, in particular in the non-conserved heptad repeats (Supplementary Fig. 5). Analysis of the experimental chemical shifts confirmed the dynamic, predominantly random coil behavior of the yCTD chain. However, a small propensity for turn formation centered at the $Y_1$ heptad position agreed best with the experimental chemical shifts (Fig. 2e). The combined data demonstrate that both structure and dynamics are replicated across the canonical heptad repeats of CTD proteins.

## CTD conformational ensemble in the dilute phase

The NMR data demonstrate that the CTD is highly dynamic, but is prone to structural biases in its conserved heptad repeats. We then analyzed the sequence-specifically assigned NOE contacts in 3R-CTD because the above comparison showed that 3R-CTD replicates the local structure and dynamics of conserved heptad repeats in hCTD/yCTD. We detected medium-range NOEs of Tyr-8 ($Y_1$ position of second heptad repeat) and Tyr-15 ($Y_1$ position of third heptad repeat) with the proline that precedes the tyrosine, e.g. Pro-6 from repeat one, as well as the succeeding proline (Pro10/17) within the same repeat (Supplementary Fig. 4d). In addition, we identified contacts between Tyr-15 and Thr-18, as well as medium-range Ser-Thr and Pro-Ser contacts (Supplementary Fig. 4d). The contacts from Tyr to the Pro in the preceding repeat suggests that the minimal structural CTD unit comprises two heptad repeats with the core structure formed by $P_{-1}S_0Y_1S_2P_3$.

Next, we subjected the experimental chemical shifts, NOEs and residual dipolar couplings of 3R-CTD to structure calculations using Rosetta[21,22]. In addition, we performed hierarchical chain growth calculations of full-length yCTD that were biased against the experimental NMR data (Supplementary Fig. 6)[23]. While both calculations generate ensembles of conformations, Rosetta biases the calculations towards more compact states, while hierarchical chain growth favors broader, more extended ensembles. The NMR-biased ensembles fulfill the experimentally determined hydrodynamic radii (Fig. 3a, f, g)[24]. Notably, the $Y_1$ position is preferentially located in turn regions in both the 3R-CTD and the yCTD ensemble. Turn conformations of $S_{-2}P_{-1}S_0Y_1$, as

characterized by O-N distances below 5 Å, were present in ~16 % of conformers. In addition, Tyr and Pro engage in multiple CH-pi contacts in the two CTD ensembles (Fig. 3b, e).

### Tyrosine interactions promote CTD phase separation

The NMR data of hCTD/yCTD in both the dilute and condensate state (Figs. 1d, 2c, 3) suggest an important role of the tyrosine $Y_1$ position in the conserved heptad repeats for CTD structure and phase separation. To validate this role, we prepared a mutant hCTD protein in which all $Y_1$ positions were replaced by phenylalanine (Y1F) or leucine (Y1L), modulating the hydrophobicity of the position 1 residue (Fig. 4a, GRAVY score[25]). In micrographs, we observed a large number of droplets for the Y1F variant, similar to the wild-type hCTD (Fig. 4a). The fluorescence recovery kinetics were also similar for wild-type and Y1F hCTD (Fig. 4b), indicating that the diffusivity inside the droplets were not perturbed by the mutation. In contrast, the replacement by leucine abolished phase separation (Fig. 4a).

To gain insight into the importance of multivalency for CTD phase separation, we next modified the frequency and distribution of tyrosine in yCTD[26,27]. We prepared three different yCTD variant proteins, in which either the N- or C-terminal 13 Tyr were replaced by Ser, or every second Tyr (Y1S CTD variants; Fig. 4c). NMR-derived hydrodynamic radii pointed to an expansion of the conformational ensemble in the dilute state when either the N- or C-terminal 13 Tyr residues were mutated (Fig. 4d). In contrast, a uniform distribution of 13 Tyr residues did not induce a strong change when compared to the wild-type protein. A uniform distribution of Tyr in the CTD sequence thus favors the compaction of the CTD ensemble in the non-phase separated state.

We then subjected the three Y1S CTD variants to phase separation assays. In microscopy experiments, no droplets were observed. However, at 100 μM we detected oligomeric particles with a diameter of ~25–150 nm by dynamic light scattering (Fig. 4e). Notably, oligomeric particles were present from 5 to 45 °C and at both 150 and 1000 mM NaCl (Fig. 4e). The mutant proteins do not phase separate at room temperature into micrometer-sized droplets at 100 μM with 5% w/v

dextran, in contrast to wild-type yCTD (Fig. 4f). The data indicate that multivalent interactions involving more than 13 tyrosine residues are required to induce CTD phase separation. With fewer tyrosines, oligomerization occurs but not droplet formation. Collectively, the experiments demonstrate that both the distribution and the number of tyrosine residues are important for the structure and phase separation of CTD.

To provide further analysis of the contribution of the CTD amino acid sequence to the protein's ability to phase separate, we prepared two designed CTD variants (YPSTSSP named PYP, and YSTPPSS named TPPS; Fig. 5a). The two variants have the same amino acid composition, but with different proximity of proline to tyrosine. For TPPS, prolines in the canonical heptad in positions 3 and 6 are swapped with residues in positions 4 and 5 (Thr and Ser), respectively, producing the new heptad YSTPPSS. Similarly, the heptad of the PYP variant interchanges the proline residues in positions 3 and 5 with positions 2 and 7 (Ser) resulting in the heptad YPSTSSP. DLS and fluorescence microscopy showed that the two CTD variants phase separate at 150 mM NaCl with increasing temperature in contrast to wild-type CTD (Supplementary Fig. 7). In addition, they form more and/or larger droplets at 500 mM NaCl, in particular at 25 °C. At 1000 mM NaCl, i.e. at very high ionic strength, both variants phase separate at 15 °C in contrast to wild-type yCTD. Additionally, larger droplets were observed by fluorescence microscopy at room temperature for the PYP variant (Supplementary Fig. 7). Probing the diffusivity of droplets formed by the three proteins using fluorescence recovery after photobleaching (FRAP) showed similar fluorescence recovery rates for yCTD and the TPPS variant, while the PYP variant displayed decreased diffusivity (Supplementary Fig. 7d). The data demonstrate that the specific sequence of amino acids in the heptad repeat influences CTD's ability to phase separate into liquid-like droplets, and affect their molecular properties.

We then analyzed by NMR intramolecular contacts in the dilute state of the two variants and compared them to wild-type CTD. To this end, we recorded two-dimensional $^1$H-$^1$H NOESY spectra and analyzed the signal intensities of the cross peaks involving aromatic tyrosine protons (Fig. 5b–d). For both CTD variants, lower cross-peak

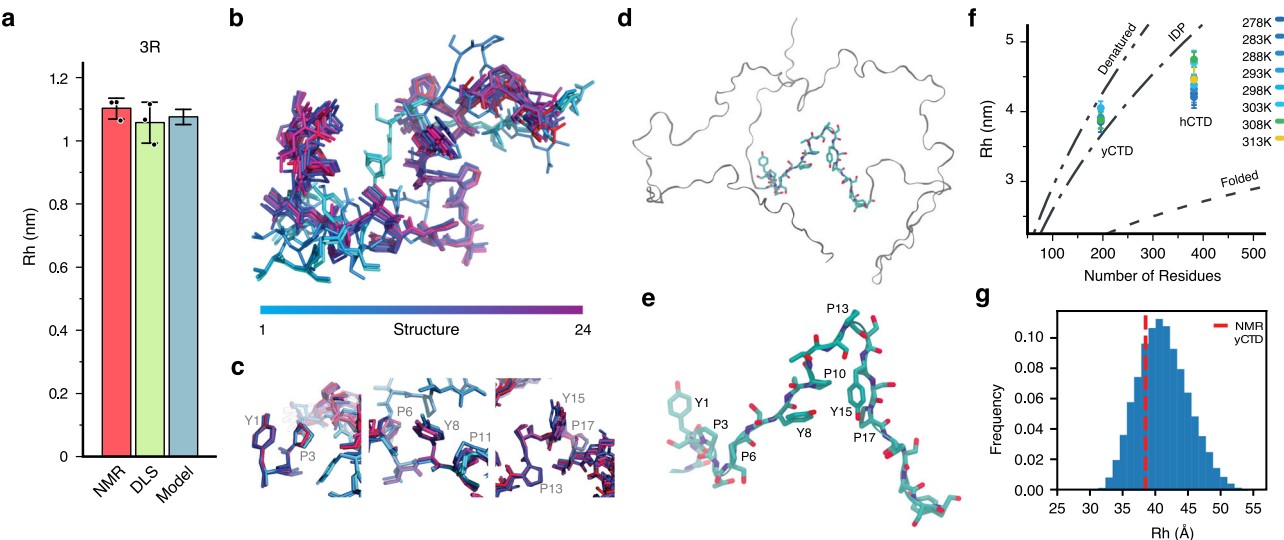

**Fig. 3 | CTD structure in the dilute phase. a** Hydrodynamic radius $R_h$ of 3R-CTD from NMR, DLS, and the ensemble of structures shown in (**b**). Error bars in panel (**a**) represent two times std. **b, c** Ensemble of low energy structures of 3R-CTD calculated with Rosetta using NMR restraints. Individual structures are colored from blue to purple. Contacts between Pro and Tyr residues are highlighted in (**c**). **d, e** Selected structure of yCTD from the ensemble of yCTD conformations generated by hierarchical chain growth (HCG) with the help of NMR data. A 21-residue fragment comprising three conserved heptad repeats is shown with side-chains.

Residue numbering in (**e**) starts with the N-terminal Tyr of the 21-residue fragment. **f** Hydrodynamic radii $R_h$ of hCTD and yCTD at increasing temperatures in the dilute phase (25 μM concentration of hCTD/yCTD). Error bars represent two times std for independent NMR diffusion measurements ($n = 3$). The curves describe the predicted tendency of $R_h$ as a function of the number of residues for fully denatured, intrinsically disordered (IDP), and folded proteins[24]. **g** Histogram distribution of $R_h$ values for the HCG structures of yCTD compared with the experimental value at 5 °C (red dashed line).

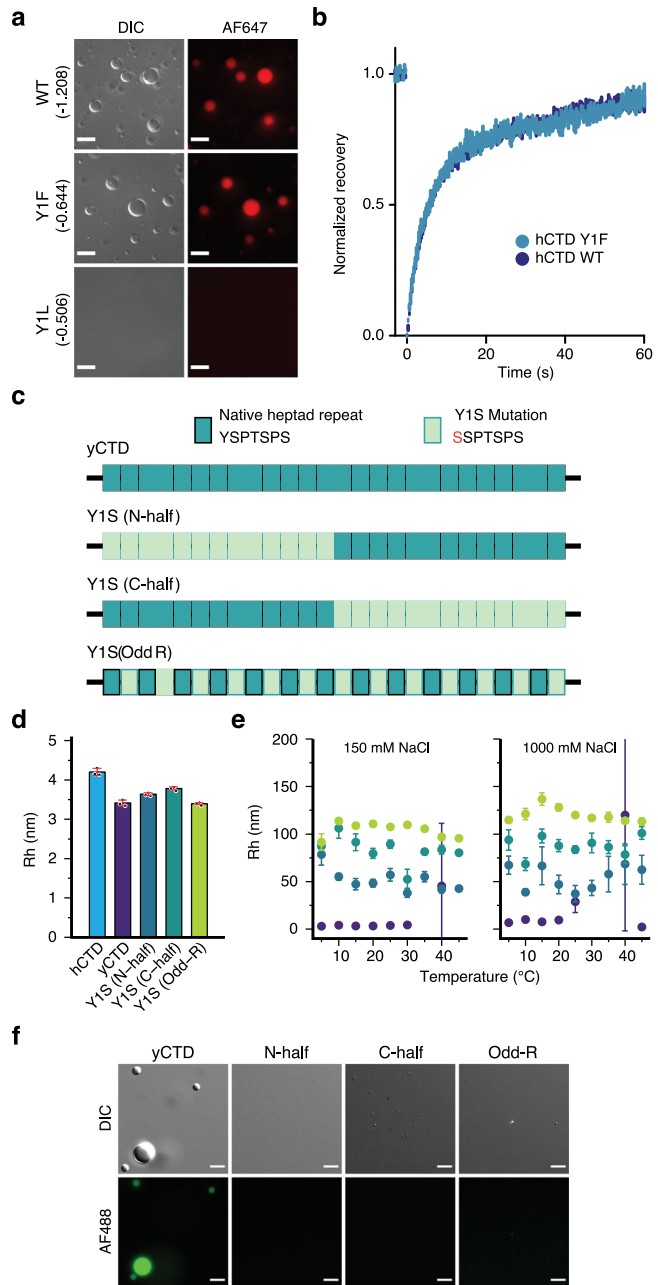

**Fig. 4 | Contribution of tyrosine residues to CTD phase separation.**
**a** Micrographs of wild-type (WT) hCTD and variants in which all Tyr residues were replaced by phenylalanine (Y1F) or leucine (Y1L). Wild-type CTD and the Y1F variant, but not the Y1L variant, form droplets at a concentration of 20 μM in the presence of 16% w/v dextran. Scale bar, 10 μm. The GRAVY score[25], indicating the hydrophobic character of each construct, is shown in parenthesis; a higher value indicates stronger hydrophobicity. **b** Superposition of the fluorescence recovery curves (n = 5) of wild-type hCTD (blue) and the variant Y1F (cyan). Curves show the average normalized recovery (mean ± standard error). **c**–**f** Influence of the distribution of Tyr residues on yCTD phase separation. Three different variants were analyzed, in which either the Tyr residues in the N-terminal half (Y1S (N-half)) or the C-terminal half (Y1S (C-half)) or every second Tyr (Y1S (Odd-R)) were replaced by Ser (schematically shown in (**c**)). Panel (**d**) compares the mean (n = 3) hydrodynamic radii of the three constructs with wild-type yCTD as determined by diffusion NMR in the dilute phase (5 °C; protein concentration 100 μM). Error bars in (**d**) represent two times std. Hydrodynamic radii of Y1S variants of yCTD in phase separation-promoting conditions (100 μM each and pH 7.4) for two different NaCl concentrations are shown in (**e**) (mean ± std). Wild-type yCTD (purple) starts to form droplets at >25 °C. Error bars in (**e**) represent std for independent measurements (n = 3). Variants in the columns of panel (**f**) were fluorescently labeled with Alexa Fluor 488 (AF488) and tested for phase separation by microscopy at similar conditions as shown in panel (**e**) (150 mM NaCl). Scale bar, 5 μm. Micrographs in panel (**a**) and (**f**) are representative of 3 independent biological replicates.

the last 0.3 μs of each MD trajectory, where radii of gyration (Supplementary Fig. 8a, b) and the average number of interacting partners per conformer in multi-copy simulations (Supplementary Fig. 8c) have reached locally equilibrated values. Stable Pro-Tyr pairs, defined as exhibiting direct van der Waals contacts for more than 10 % of simulated time, tend to form tight configurations, whereby Pro and Tyr rings are oriented either in a parallel, stacked conformation or orthogonally to each other. Such configurations correspond to the distances of ~4 Å between the rings' centers of geometry (peaks of the distributions, Fig. 6b). In the case of intramolecular contacts, the sequence-neighboring Pro-Tyr pairs in the canonical heptad also populate the second peak around 6–7 Å of the corresponding distributions (Fig. 6b). Notably, hCTD in the multi-copy, dense-phase simulations adopts with an appreciable frequency configurations that resemble those of the Rosetta-based 3R-CTD ensemble in the dilute phase (Supplementary Fig. 8d).

Overall, the predominant configurations of the interacting Pro-Tyr pairs correspond to a stacked configuration of the two rings as shown in Fig. 6c for the RMSD cut-off for clustering of 0.7 Å. Expectedly, the population of top structural clusters depends on the applied RMSD cut-off (Fig. 6d). With small cutoff values, the analysis is very discriminatory and results in low populations of the top clusters. Populations increase with an increase in the cut-off, reaching 70 % and 52 % in the intra- and intermolecular contexts at the cutoff of 0.1 nm, respectively. The top clusters in the latter case comprise all states around the main peaks in the distance distribution (Fig. 6b).

The preference for forming tightly interacting pairs results in high fractions of Pro-Tyr contacts in the pool of all contacts detected between interacting hCTD molecules (Fig. 7a). Over the last 0.3 μs of MD simulations, the frequency of Pro-Tyr contacts (14 %) reaches the same level as the frequency of Pro-Ser contacts, which based on the hCTD sequence composition are expected to be the most frequent (Fig. 7a). Further analysis showed that in dense-phase simulations intermolecular Pro-Tyr contacts are substantially enriched over the randomized background (enrichment of 1.8 x), while the Pro-Ser contacts are depleted (enrichment of 0.7 x) (Supplementary Fig. 8e), and similarly so in dilute phase simulations, albeit less pronounced (Supplementary Fig. 8f).

The interaction patterns for heptad positions differ between intra- and intermolecular contexts (Fig. 7b). Within a single hCTD molecule, the interactions between neighboring positions (along the diagonal)

intensities were present for the epsilon position of the tyrosine ring (Fig. 5b, c). Additionally, the TPPS variant showed lower cross-peak intensities for the delta position (Fig. 5d). The combined observation of enhanced phase separation and reduced intramolecular tyrosine contacts, including Tyr-Pro contacts, of the two sequence-perturbed variants suggests that intramolecular contacts involving the aromatic ring of tyrosine compete with intermolecular contacts driving CTD phase separation.

### Proline-tyrosine contacts are enriched upon crowding

Next, we performed all-atom 1 μs-long molecular dynamics (MD) simulations in explicit solvent to study the molecular interactions determining CTD phase separation. We performed independent simulations for hCTD at high dilution (single-copy system) and in a crowded context (multi-copy system with 10 copies of hCTD in the simulation box, Fig. 6a, Supplementary Fig. 8a, c). Subsequently, we carried out a detailed analysis of spatial configurations of Pro-Tyr interacting pairs and the overall statistics of contact formation over

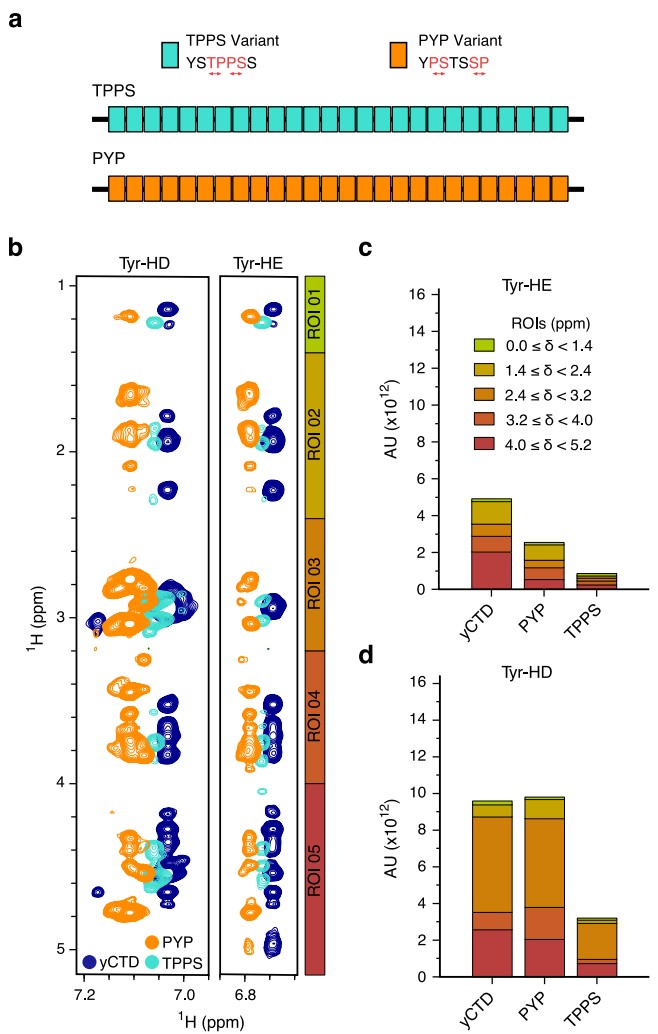

**Fig. 5 | Aromatic and side-chain intramolecular contacts. a** The design variants TPPS and PYP with the same amino acid composition as yCTD but changed sequence. Swapped residues are shown in red. **b** Overlay of the aromatic regions of two-dimensional ¹H-¹H NOESY spectra of yCTD and its PYP and TPPS variants recorded in the dilute phase. Five regions of interest (ROI) were defined and used for signal integration. **c, d** Integrals of the NOE peaks (arbitrary units; AU) extracted from the spectra in panel (**b**). The integrals are classified into two vertical categories corresponding to the chemical shift of the aromatic protons in positions epsilon (**c**) and delta (**d**). The stacks of each graph are divided into the five horizontal regions of chemical shifts (δ; vertical scale) indicated graphically in (**b**).

dominate, pointing to a local character of hCTD structural organization. In between different hCTD molecules, on the other hand, Tyr represents the most interacting residue, with preferred partners being either other Tyr residues or proline at position 6 (Pro-6). According to an analysis of the sequence composition of all heptad repeats in hCTD, Pro-6 is most conserved (Fig. 7a, bottom). For both Pro residues in the hCTD heptad, Tyr is the most preferred interaction partner. The sequence-specific analysis further showed that the high propensity of Tyr towards interaction is distributed along the hCTD sequence in the multi-copy system with a slight preference for the more conserved, N-terminal heptad repeats (Fig. 7c).

Next, we analyzed inter- and intramolecular contacts in MD simulations of two other low-complexity protein regions, namely the intrinsically disordered regions of LGE1 and Fused in Sarcoma (FUS). The MD simulations of LGE1 and FUS were performed using the same force field parameters and water model as with hCTD[28]. For both LGE1 and FUS, Pro-Tyr contacts are more enriched and populated in

between molecules than within a single molecule (Fig. 7d). The strongest enrichment of intermolecular Pro-Tyr contacts is observed for FUS (Fig. 7d). The analysis suggests that Pro-Tyr contacts may more broadly contribute to phase separation and condensation of intrinsically disordered proteins.

**Associative phase separation with the human Mediator complex**
Next, we prepared the 1.37 MDa human Mediator complex (hMED)[29–32] to investigate co-recruitment between hMED and hCTD. Part of the hMed sample was fluorescently labeled with Alexa Flour 647. We subjected the hMED complex alone and in the presence of 5 % w/v of the molecular crowder dextran to phase separation experiments. Without dextran, no droplet-like structures were observed at 500 nM hMED by fluorescence microscopy (Fig. 8a). In contrast, hMED-containing droplets were abundant in the presence of dextran (Fig. 8a). The human Mediator complex thus undergoes phase separation at submicromolar concentrations in crowded conditions.

We then added 5 µM of hCTD to 500 nM hMED solutions (Fig. 8a). At this condition, hCTD does not phase separate alone (Supplementary Fig. 9a). Instead, we observed that hCTD is concentrated inside hMED-containing droplets. Further phase separation experiments confirmed the co-recruitment of hCTD and hMED into condensates. Above 50 µM, hCTD phase separates into droplets without requiring dextran (Supplementary Fig. 9b). When 500 nM of hMED are added, hMED concentrates inside the hCTD droplets (Supplementary Fig. 9c; Fig. 8a). The data demonstrate that hCTD and hMED can phase separate together in vitro, in agreement with experiments in cells[10,18].

To probe the importance of the tyrosine residues in the conserved heptad repeats for CTD/hMED recruitment, we made use of 100 µM yCTD and its variants. We then added 500 nM of hMED to the samples. Microscopy revealed a mixture of droplets enriched in both wild-type yCTD and hMED (Supplementary Fig. 9d). In contrast, the phase-separation impaired Tyr-to-Ser yCTD variants co-localized less with hMED condensates, but not the TPPS and PYP yCTD variants (Supplementary Fig. 9d). Finally, the reduction in the apparent diffusion of hCTD in hCTD/hMED droplets at equimolar ratio suggests restricted hCTD mobility through interaction with hMED (Supplementary Fig. 9e, f).

Insights into possible molecular interactions determining combined phase separation of CTD and hMED can be derived from the structure of the pre-initiation complex in which short stretches of the CTD are resolved bound to hMED (Fig. 8b, e). Pro-Tyr, Pro-Pro, and Tyr-Tyr contacts are present between the CTD fragments of RPB1 and hMED (Fig. 8c, e). In addition, some of the hMED-bound heptad repeat structures can be found in the experimentally determined conformational ensemble of 3R-CTD (Supplementary Fig. 10), suggesting that the hMED-bound states of the heptad repeats are transiently performed in solution. Although it is currently not known whether such stable contacts can occur in CTD/hMED condensates, our experiments and analyses suggest that molecular interactions involving tyrosine and proline are important for combined condensation of Mediator and CTD, and thus Pol II.

## Discussion
CTD-mediated phase separation of RNA polymerase II provides a simple mechanism for gene activation[8,14,33]. In this model, CTD–CTD interactions cluster unphosphorylated Pol II into nucleoplasmic hubs. When the hubs are proximal to gene promoters, high concentrations of Pol II in the hubs can enable high initiation rates during activated transcription[14,33]. CTD–CTD interactions thus may be critical for gene transcription in eukaryotic cells. The molecular determinants of CTD–CTD interactions and phase separation have, however, been largely unknown. Here, we showed that human CTD, as well as yeast CTD, phase separate alone without co-factors or molecular crowding agents at physiological temperature. The tyrosines of the canonical

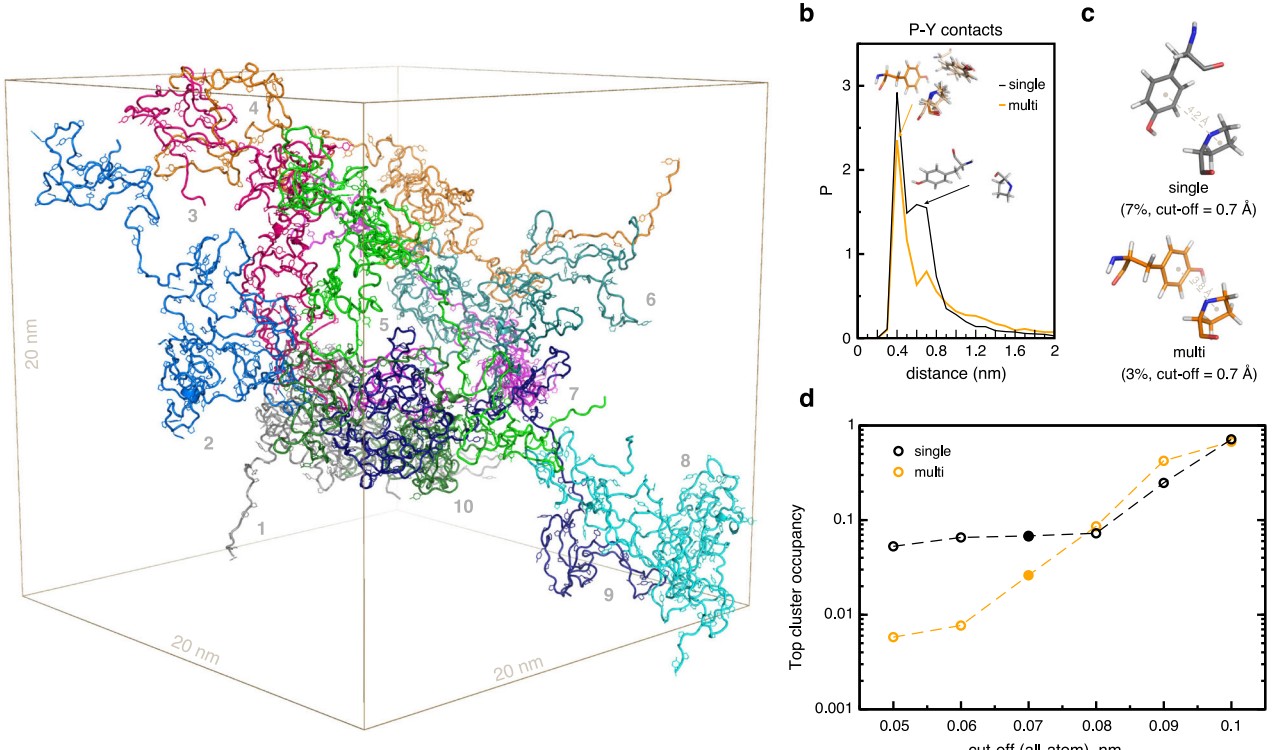

**Fig. 6 | Analysis of intermolecular tyrosine-proline interactions in MD simulations. a** Visualization of the MD simulation box for the multi-copy system (10 protein copies). Each protein copy is colored differently and is labeled with a number. Proteins are shown in cartoon representation with Pro and Tyr side-chains in stick representation. **b** Distribution of distances between Pro and Tyr rings in a subset of Pro and Tyr pairs with contact frequency above 10%. Representative configurations (central structures of structural clusters) of Tyr-Pro pairs in single-copy (single) and multi-copy (multi) systems are shown in relation to the actual distance between the residues. **c** Representative configurations of Tyr-Pro pairs for the most highly populated (top) structural clusters with the applied all-atom RMSD cut-off of 0.7 Å. The shown structures correspond to the filled circles in (**d**). Cluster populations and distances between the residues are indicated. **d** Occupancy of the top structural clusters with respect to the applied all-atom RMSD cut-off used for clustering. The populations of the top clusters were estimated relative to the total number of configurations in joint master Tyr-Pro MD trajectories (see Methods) with a separation distance <2 nm (see (**b**)).

$Y_1S_2P_3T_4S_5P_6S_7$ heptad repeat sequence of CTD engage in intra- and intermolecular interactions that shape CTD structure and phase separation[34]. NMR spectroscopy and molecular simulations show that favorable interactions between the aromatic rings of tyrosine and the other residues of the canonical heptad repeat are abundant in the CTD. Contacts that are present in the condensed phase of CTD include Tyr-Pro interactions. Intermolecular Tyr-Pro interactions are also observed in MD simulations of the crowded phases of other low-complexity proteins. Additionally, co-recruitment of the human Mediator complex and CTD during phase separation suggests that Tyr-Tyr interactions are important for multi-component condensed phases of Pol II and transcriptional activators.

Despite its importance for gene regulation, the structure of the CTD has remained largely enigmatic. The low-complexity $Y_1S_2P_3T_4S_5P_6S_7$ heptad repeats of CTD impart a dynamic conformational ensemble[35]. A further challenge is provided by the repetitive nature of the CTD sequence. Early work has therefore focused on short CTD peptides, sometimes circularized and often at low pH in turn-promoting solvents to stabilize structure[36,37]. The structure of short CTD fragments in complex with CTD-binding partners has also been determined[38,39]. In addition, the structural properties of non-repetitive regions of the *Drosophila melanogaster* CTD have been characterized[40,41]. Using a combination of NMR spectroscopy and structure calculations we here determined conformational ensembles that describe the dynamic structure of the canonical CTD heptad repeats in both CTD peptides and yCTD. The core structuring element in these ensembles is formed by the sequence $P_{-1}S_0Y_1S_2P_3$ at the interface between two canonical

repeats. NMR analysis further suggests an identical conformational sampling of the canonical heptad repeats in hCTD.

We showed that pure and tag-free human CTD phase separates alone without crowding agents (Fig. 1). CTD phase separation depends on CTD concentration, occurs above a lower critical temperature, and does not require phosphorylation (Fig. 1). The high density and uniform distribution of tyrosine residues in the CTD sequence is important for CTD phase separation (Fig. 4). Notably, substitution of tyrosine for phenylalanine in the low-complexity domains of the proteins Fused in Sarcoma and LAF-1 attenuates LLPS[42]. In contrast, replacement of tyrosine by phenylalanine in hCTD had little influence on the protein's ability to form droplets (Fig. 4a, b). The data suggest that in case of hCTD phase separation, CH-pi and pi-pi interactions maybe more important than hydrogen bond formation for intermolecular association.

Phase separation of proteins with low-complexity regions often depends on multivalent interactions among tyrosine residues from prion-like domains and arginine residues from RNA-binding domains[43]. Using atomistic molecular dynamics simulations, we showed that Tyr-Pro interactions – together with other intra- and intermolecular interactions – play an important role in the condensed phase of CTD (Figs. 6, 7): the negatively charged π face of the aromatic ring of tyrosine interacts with the partially positively charged ring of proline[44]. While local interactions dominate in the dilute phase, intermolecular Tyr-Pro contacts between CTD molecules are present in the condensed phase (Figs. 6, 7a, c). We also observed intermolecular Tyr-Pro contacts in other low-complexity proteins (Fig. 7d), suggesting a broader role of Tyr-Pro interactions in the phase separation of low-complexity proteins.

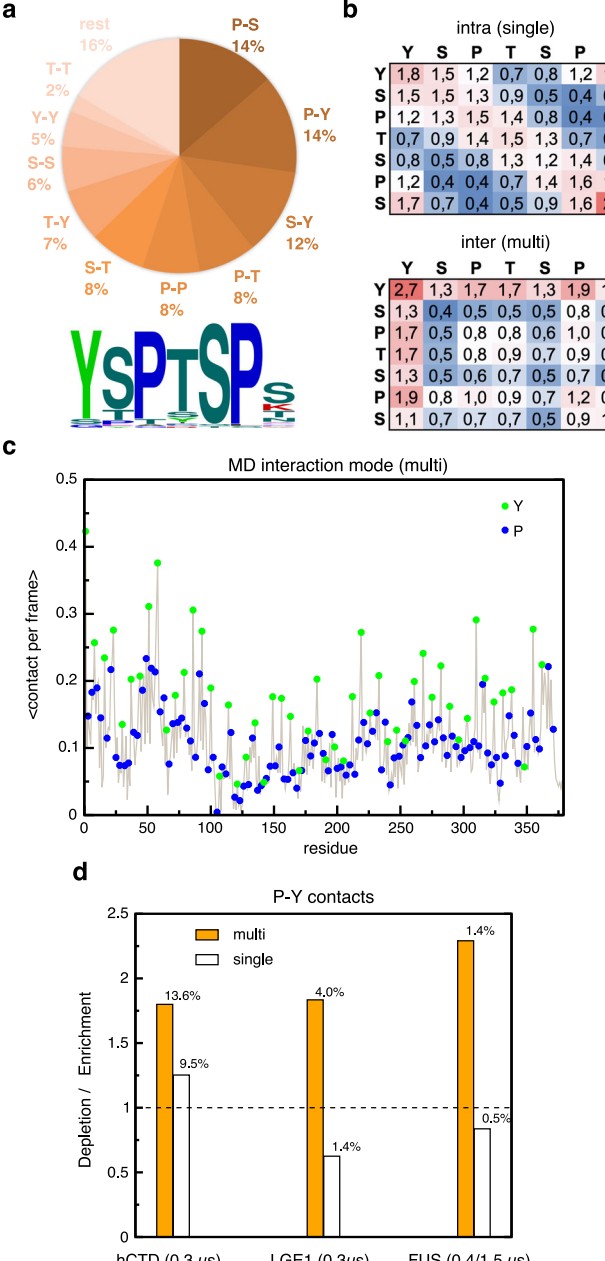

**Fig. 7 | Enrichment of intermolecular tyrosine-proline interactions in simulated crowded environments. a** Absolute fractions of the top 10 residue-residue contacts in the multi-copy system. Below, the sequence logo corresponding to the composition of heptads in the full-length hCTD sequence is shown. **b** Position-resolved interaction matrix in hCTD heptads obtained for single-copy (intramolecular contacts) and multi-copy (intermolecular contacts) systems. A value given for each pair corresponds to the ratio of the frequency of a particular contact type in the pool of all contacts seen in MD and the same frequency in a randomized background. Contacts colored red are enriched, and those shown in blue are depleted. Only positions corresponding to the canonical heptad residues are considered for the analysis. **c** Distribution of intermolecular contact frequencies along the hCTD sequence, averaged over the 10 protein copies in multi-copy simulations. Sequence positions corresponding to Tyr and Pro residues are indicated with green and blue-filled circles, respectively. **d** Comparison of depletion/enrichment values for Tyr-Pro contacts in the inter molecular context (multi-copy systems) to those in the intra-molecular context (single-copy systems) estimated as a ratio between the observed and the expected fractions of contacts, with the latter being evaluated from the frequency of residues in question. The corresponding fractions of the contacts are indicated above bars. The results of hCTD simulations are shown in comparison to the statistics obtained for other disordered low-complexity proteins (LGE1[28], FUS), simulated using the same modeling framework.

Transcriptional activators form condensates near enhancers[12,45]. Condensates of transcriptional activators may recruit Pol II[10,18]. Additionally, transcriptional activators may assist in Pol II hub formation when Pol II concentration is subcritical[33]. The Pol II CTD physically interacts with Mediator, which functions as a transcriptional coactivator in eukaryotes[46,47]. Consistent with the formation of multi-component Pol II/Mediator condensates[10,18], we showed that the purified 1.37 MDa human Mediator complex is recruited into in vitro droplets of human CTD (Fig. 8). In addition, the Mediator complex phase separated into droplets at sub-micromolar concentration in crowded conditions into which CTD was recruited (Fig. 8). We also showed that CTD's tyrosine residues are important for the formation of this multi-component condensates, in agreement with abundant Tyr-Pro, Tyr-Tyr and Pro-Pro contacts between CTD and Mediator in the structure of the Mediator-bound preinitiation complex (Fig. 8)[30]. Tyr-Pro, Tyr-Tyr and Pro-Pro interactions may thus contribute to different multi-component condensed phases of Pol II. For example, CTD can interact with condensates of FET (FUS−EWS−TAF15) proteins as well as with the splicing factors SRSF1/SRSF2[18,48]. Other multivalent interactions will also contribute to the formation of multi-component condensed phases of Pol II, in particular from the less conserved distal part of human CTD which contains lysine residues[48–50].

Post-translational modification of the CTD repeats is intimately connected to eukaryotic gene transcription[51]. An unphosphorylated CTD is necessary for the assembly of the pre-initiation complex at Pol II promoters[52]. The transition of Pol II into active elongation is subsequently stimulated by phosphorylation at $S_5$ in the canonical heptad repeats[52]. Previous studies found that $S_5$-phosphorylation induces sequence-specific conformational switches in the CTD and slightly expands its conformational ensemble[40,41,53,54]. Notably, $S_5$-phosphorylation by the transcription initiation factor IIH kinase CDK7 dissolves CTD droplets providing a mechanism for promoter escape and transcription elongation[14]. CDK7-phosphorylated CTD may then engage into other transcriptional condensates such as those formed by the positive transcription elongation factor b (P-TEFb) or splicing factors[8,18]. The importance of interactions involving tyrosine for CTD structure and phase separation shown in the current study, however, emphasizes the need for further studies investigating the role of tyrosine phosphorylation in modulating Pol II condensation. CTD tyrosine phosphorylation impairs termination factor recruitment to RNA polymerase II and controls global termination of gene transcription in mammals[55–57]. Protein factors such as prolyl isomerases as well as nucleic acids may provide a further level of regulation of the CTD-mediated condensation of RNA polymerase II in eukaryotic gene transcription.

## Methods

### CTD expression and purification

Plasmids were modified from the original construct to produce the carboxyl-terminal domain of human Pol II (hCTD; RPB1 residues 1593–1970) described previously[14]. The constructs for hCTD, its variants (Y1F & Y1L), yCTD, and the yCTD variants (Y1S mutants, TPPS, and PYP) are composed of histidine (6xHis) and maltose binding protein (MBP) tags located at the N-terminus. A flexible linker of ten consecutive asparagines and the tobacco etch virus (TEV) protease cleavage site were introduced to allow cleavage of the tags. The protein sequences were codon-optimized for expression in bacteria (GenScript). For site-specific labeling with a fluorescent tag, a cysteine residue was present at the N-terminus downstream of the TEV cleavage.

MBP-tagged proteins were overexpressed in *E. coli* BL21 RP-Codon Plus DE3 cells (Agilent Cat. #230255) at 37 °C in LB media. M9 media was used for the production of $^{15}$N and $^{15}$N/$^{13}$C-labeled proteins, and overexpression was achieved according to Marley et al.[58]. Media were supplemented with ISOGRO (Sigma) and selected isotopes.

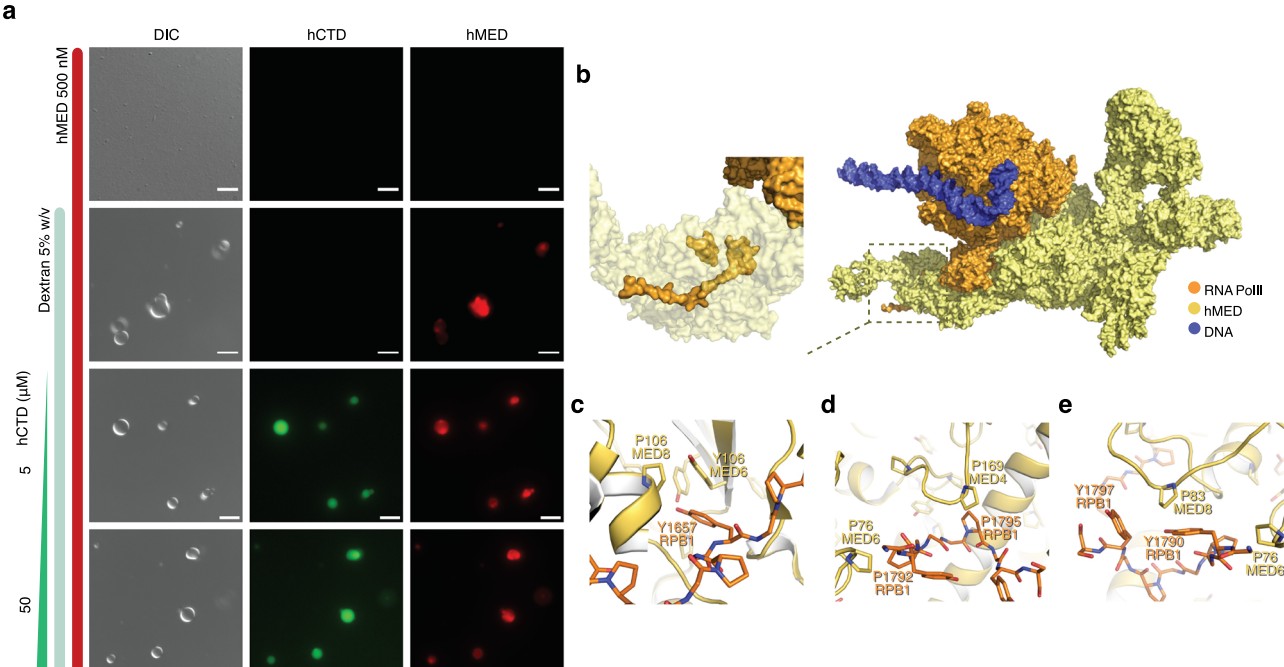

**Fig. 8 | Co-recruitment of Mediator complex and human CTD into condensates.**
**a** Differential interference contrast and fluorescence microscopy of the phase separation of hCTD and the human Mediator complex. Different mixtures and conditions are indicated with vertical bars. hCTD was labeled with Alexa Flour 488 (AF488; green), and the human mediator complex hMED with Alexa Flour 647 (red). **b**–**e** Structure of the mediator-bound preinitiation complex (PDB id 7ENC[25]). Some proteins were omitted for better visualization in panel (**b**). The structure of the hMED-bound CTD is displayed in (b, inset). Panels **c**–**e** highlight Tyr-Tyr (**c**), Pro-Pro (**d**), and Tyr-Pro (**c** and **e**) contacts between mediator subunits and hCTD. Micrographs are representative of 3 independent biological replicates. Scale bar, 5 μm.

Cells were collected after overexpression (10 min, 10000 x *g*; Avanti JXN-26, Beckman Coulter) and resuspended in lysis buffer (25 mM HEPES, pH 7.4, 300 mM NaCl, 30 mM imidazole, cOmplete EDTA-free protease-inhibitor cocktail, 0.1 mg/l lysozyme and 0.1 mM PMSF) at 4 °C. Cells were disrupted by sonication (15 s pulse at 60 W, 45 s pause, 10 min total; SONOPULS, Bandelin). The cell extract was clarified by centrifugation (20 min, 45000 x *g*; Avanti JXN-26, Beckman Coulter), loaded with a sample pump (Äkta Pure GE Healthcare) into an ion-metal affinity chromatography column (IMAC; FastFlow-Hitrap GE Healthcare), and eluted with imidazole. The purity of the fractions was improved by size exclusion chromatography (Superdex 75 26/600 GE Healthcare). Fractions containing the MBP-tagged protein were merged and concentrated. TEV protease was added (1:100 mass ratio) to the mix for cleavage. The reaction was incubated overnight (16-18 hours) at 4 °C with gentle agitation. Cut tags were removed using IMAC purification, collecting and concentrating the unretained fractions. Fast protein liquid chromatography (FPLC) was performed at 4 °C using an Äkta Pure system (GE Healthcare). Purified protein was collected from Reversed-Phase HPLC (preparative column: Vydac 214TP 5 μm C4, 250 x 10 mm; A: water + 0.1% TFA, B: acetonitrile + 0.1% TFA; HPLC system JASCO with diode array detector) and the molecular weights were confirmed by mass spectrometry (analytical column: Waters BioResolve RP mAb, Polyphenyl, 450 A, 2,7 m, 4.6 x 100 mm; A: water + 0.1% TFA, B: acetonitrile + 0.1%TFA; LC-MS: Acquity Arc System, Waters, with SQD2-Mass-Detector: Single Quadrupole; Direct mass: ZQ 4000 Waters, Single Quadrupole, injection by syringe pump). HPLC samples were lyophilized for further experiments. For the variants Y1F and Y1L, the HPLC purification was not performed. In this case, protein concentration was determined based on the predicted molar extinction coefficient after purification with a Superdex 200 10/300 Increase column (GE Healthcare). Concentrated protein solutions (>100 μM) were divided into small aliquots (5-10 μL), frozen in liquid N₂, and stored at −80 °C until further use.

1R-CTD, 2R-CTD, 3R-CTD, 4R-CTD, and 6R-CTD peptides were synthesized by GenScript and carried acetyl-protection groups at the N-terminus.

### Human mediator complex production/purification
The 26-subunit containing, human Mediator complex (MW = 1.37 MDa) was expressed and purified from *Spodoptera frugiperda* cells. Four different constructs (C1-C4) for baculovirus expression were used. The C1-C3 constructs were described previously[31] with one exception, the addition of the MED1 subunit to C2. To generate the C4 construct, MED15, MED16, MED24, N-terminal maltose binding protein (MBP) tagged MED25 and C-terminal MBP tagged MED23 were incorporated into a modified pFastBac vector using ligation-independent cloning[59].

Bacmid preparation and virus production were performed as described previously[60]. Expression of Mediator in insect cells was achieved by co-infection of the V1 virus for constructs C1-C4 in *Sf*21 cells After 48-60 h of expression, cells were collected by centrifugation (900 x *g*, 10 min, 4 °C) and resuspended in Buffer A (20 mM 4-(2-hydroxyethyl)-1-piperazineethanesulfonic acid (HEPES) pH 7.5, 300 mM NaCl, 10% glycerol (v/v), 0.5 mM Tris(2-carboxyethyl)phosphine (TCEP), 0.284 μg/ml leupeptin, 1.37 μg/ml pepstatin A, 0.17 mg/ml PMSF and 0.33 mg/ml benzamidine). The cell suspension was flash-frozen in liquid nitrogen and stored at −80 °C.

All protein purification steps were performed at 4 °C unless otherwise stated. Recombinant Mediator was purified by affinity chromatography followed by size-exclusion chromatography (SEC). Stored insect cell suspension was thawed in a water bath at 25 °C. Cells were lysed by sonication and clarified by centrifugation (79000 x *g*, 60 min). Filtered supernatant was passed over amylose resin pre-equilibrated with Buffer A and then washed with 40 column volumes (CV) of the same buffer. Mediator was eluted with Buffer B (20 mM HEPES pH 7.5, 300 mM NaCl, 10% glycerol (v/v) and 1 mM mM TCEP) containing 100 mM maltose and incubated overnight with TEV protease. The cleaved MBP tag, TEV protease and excess Mediator

subunits were removed by SEC over a Superose 6 increase 10/300 GL column (Cytivia) equilibrated with Buffer C (20 mM HEPES pH 7.4, 300 mM NaCl, 10% glycerol (v/v) and 0.5 mM TCEP).

Fractions were analysed by SDS–PAGE and the homogenous peak fractions were pooled and concentrated to between 4.5–5 mg/ml using a 100-kDa MWCO Amicon Ultra Centrifugal Filter (Merck). The presence of all 26 Mediator subunits in the pooled fraction was confirmed by mass spectrometry analysis. Concentrated Mediator was aliquoted, flash-frozen in liquid nitrogen and stored at −80 °C until use.

## Phase separation assays
Stock solutions (200 μM) were produced by weighting dry protein and dissolving it in pre-cooled buffers with 50 mM NaCl at 4 °C (to avoid initial phase separation). Buffer solutions were filtered (0.22 μm) after preparation to avoid interference from impurities. Protein concentrations were tested from 1 μM to 100 μM at pH values of 6.2 (25 mM; MES) and 7.4 (25 mM; HEPES) and different ionic strengths (50, 100, 150, 500 and 1000 mM NaCl). When indicated, dextran T400 (Pharma) was used as a crowding agent at 5 % w/v.

## Dynamic light scattering
Hydrodynamic radius ($R_h$) measurements were recorded in a DynaPro NanoStar spectrometer (Wyatt technology) equipped with a temperature control system. A weighted average mean (w.a.m.) of $R_h$ is reported at different ionic strengths; sigmoidal regression lines were added to emphasize the transition between the diluted and condensed (liquid-like droplets) states.

## Microscopy
hCTD and yCTD were labeled with Alexa Fluor 488 maleimide (AF488) according to the protocol in the microscale kit provided by the manufacturer (Invitrogen). Sub-micromolar amounts (< 0.5 μM) of fluorescently labeled protein were mixed with unlabeled protein to reach the final concentrations. Prior to imaging, samples were incubated for 5-10 minutes on ice (4 °C) and gently mixed by pipetting. Five microliters of sample were loaded onto glass slides and covered with ø18 mm coverslips. Differential interference contrast (DIC) and fluorescence micrographs were acquired at room temperature using a Leica microscope (DM6000B) equipped with a x63/1.20 objective (water immersion) and x100/1.40-0.70 objective (oil immersion).

A small portion of the human mediator complex (hMED) was labeled with Alexa Fluor 647 NHS ester (AF647; microscale kit, Invitrogen) for phase separation experiments. Protein samples were combined by pipetting and incubated in ice for two minutes in 25 mM HEPES, pH 7.4, 150 mM NaCl, 1.0 mM TCEP. In addition, dextran T400 (Pharma) was added (5% w/v) as a crowding agent when indicated. Co-recruitment was investigated at room temperature by DIC and fluorescent microscopy using a Leica DM6000B microscope. Micrographs were analyzed and processed with Fiji (NIH). Micrographs are representative of at least three independent biological replicates.

## Fluorescence recovery after photobleaching (FRAP)
Co-recruitment was investigated at room temperature mixing protein samples and fluorescently labeled samples (AF488 and AF647) by pipetting either in 25 mM HEPES, pH 7.4, 150 mM NaCl, 1.0 mM TCEP, dextran T400 (Pharma) 5% w/v (CTD variants) or 20 mM HEPES, pH 7.4, 220 mM NaCl, 1.0 mM TCEP, 16%w/v dextran (WT-CTD, Y1F and Y1L variants). Images were recorded using confocal microscopes Zeis LSM880 and Leica SP8 equiped with a 63x oil and water immersion objectives, respectively. Two iterations per bleaching on the CTD variants and single iteration on WT-CTD, Y1F and Y1L variants were used. Triplicate replica was performed on each setup for CTD variants while quintuplicate replica for WT-CTD, Y1F and Y1L variants. Protein samples were labeled as described above in the Microscopy section. Data analysis was performed in Fiji (NIH).

## Nuclear magnetic resonance
Protein samples for NMR were prepared in 25 mM sodium phosphate buffer, 50 mM sodium chloride, pH 6.2, 10% v/v D2O, and supplemented with 50 μM sodium trimethylsilylpropanesulfonate (DSS) for chemical shift referencing. NMR spectrometers (Bruker) were equipped with triple resonance cryogenic probes. Spectra were processed using NMRPipe[61]. Resonance assignments were performed using NMRFAM-SPARKY[62].

For NMR measurements of the CTD peptides, two millimolar solutions of each peptide were prepared. Two-dimensional ${}^1$H-${}^1$H TOCSY (80 ms mixing time; Bruker Avance NEO at 800 MHz), ${}^1$H-${}^1$H NOESY (80 and 25 ms mixing time; Bruker Avance NEO at 800 MHz), ${}^1$H-${}^{15}$N Heteronuclear Single Quantum Coherence (HSQC; Bruker Avance NEO at 600 MHz equipped with triple resonance prodigy probe) and ${}^1$H-${}^{13}$C HSQC experiments (Bruker Avance NEO at 800 MHz) were recorded. Peptide assignments were compared with ${}^1$H-${}^{15}$N HSQC spectra of 25 uM ${}^{15}$N-labeled hCTD acquired at 5 °C (Bruker Avance NEO operating at 1200 MHz). NMR spectra of unlabeled yCTD (120 μM) were acquired on a Bruker Avance Neo 800 MHz spectrometer. NMR spectra of ${}^{13}$C/${}^{15}$N-labeled yCTD (100 μM) were acquired at 800 MHz (Bruker Avance Neo) and 900 MHz (Bruker Avance III HD) spectrometers. In addition, NMR spectra of ${}^{13}$C/${}^{15}$N-labeled yCTD (400 μM) were acquired on Bruker 700 MHz (Bruker Avance III HD) and 900 MHz (Bruker Avance III HD) spectrometers.

Sensitivity-enhanced ${}^1$H-${}^{15}$N IPAP-HSQC experiments were recorded at 5 °C on a Bruker Avance III HD 900 MHz spectrometer for 2R-CTD and 3R-CTD peptides isotropic sample and with 30 mg/mL of the alignment media Pf1 bacteriophage for the anisotropic sample to measure residual dipolar couplings.

Hydrodynamic radius values were determined by diffusion NMR[63]. For 2R-CTD, 3R-CTD, 4R-CTD, and 6R-CTD, 1.0 mM samples were employed and measured at 5 °C on a Bruker Avance III HD spectrometer at 900 MHz. For hCTD, yCTD, and the Y1S yCTD variant proteins, 100 μM of protein concentration was used and spectra were recorded at 5-35 ˚C on Bruker Avance NEO at 800 MHz and Bruker Avance III HD at 900 MHz spectrometers.

BEST-TROSY[64] versions of the three-dimensional triple resonance experiments HNCO, HN(CA)CO, HNCACB, HN(CO)CACB, HNCA, and HN(CO)CA in combination with TROSY-(H)N(CA)NNH, TROSY-H(NCA)NNH, as well as two-dimensional ${}^1$H-${}^{15}$N HSQC, ${}^1$H-${}^{15}$N TROSY-HSQC[65–68] and ${}^1$H-${}^{13}$C HSQC spectra, were acquired for sequential backbone resonance assignment of yCTD. Non-uniform sampling was used for the three-dimensional experiments adjusting the sampling percentage (≥ 25-50 %) based on the signal-to-noise ratios for the 2D projections. Spectra were recorded at 800 MHz (Bruker Avance Neo) and 900 MHz (Bruker Avance III HD) spectrometers. Secondary structure propensities were calculated using TALOS[69,70] based on the unambiguous chemical shifts derived from the resonance assignment for yCTD and supplemented with the chemical shift values obtained for 3R-CTD for the degenerated resonances of the canonical repeats.

To detect interresidue contacts in CTD condensates, two-dimensional ${}^1$H-${}^1$H-NOESY spectra were recorded, optimizing the mixing time (20-600 ms) in the diluted and condensed conditions. The dilute state was recorded at low temperature (5 °C) with a 2.1 mM yCTD sample. The condensed condition was reached by adding 5 % w/v dextran T400 and increasing the temperature to promote phase separation. Intraresidue contacts in PYP, TPPS, and wild-type yCTD were extracted from two-dimensional ${}^1$H-${}^1$H-NOESY spectra (mixing time of 120 ms) recorded in the dilute phase. Spectra were processed using NMRPipe[61], and the volume of the peaks was quantified using NMRDraw[61].

## Rosetta structure calculations
Backbone chemical shifts (HN, N, C, Cα, and sparse Cβ), residual dipolar couplings, and NOE restrictions were used in RASREC

CS-Rosetta to calculate an ensemble of structures for 3R-CTD. Reference dihedral angles from fragments of 3-mers and 5-mers were picked using the chemical shifts along with 16 RDCs and 36 NOE restraints were manually assigned from the two-dimensional $^1$H-$^1$H-NOESY spectrum of 3R-CTD (mixing time of 250 ms). The spatial restrictions were iteratively evaluated to avoid violations. Five thousand structures were produced during calculations, further selected based on the agreement with the experimentally derived spatial restrictions from NMR and the standard Rosetta all-atom energy functions (200 models). Hydrodynamics radii of the Rosetta-derived structures of 3R-CTD were calculated using HullRad V8[71] for further filtering with the experimental value defining an ensemble of 24 models.

### Hierarchical chain growth ensembles

Conformational ensembles of yCTD were generated using reweighted hierarchical chain growth (https://github.com/bio-phys/hierarchical-chain-growth)[23,72]. For yCTD we simulated 64 fragments using replica exchange molecular dynamics. Each replica was simulated for 1.9 μs using the Amber99sb-star-ildn-q protein[73–77] and TIP3P water model[76]. We adjusted the simulation protocol[72] to sample the cis-trans equilibrium of proline residues[78], simulating 32 replicas at temperatures from 300 K to 540 K. The total explicit solvent atomistic simulation data set amounts to 3.9 ms. Exchanges between neighboring replicas were attempted every 1 ps. Replica exchange simulations were run using GROMACS[79].

For refinement of yCTD fragments against Cα, Cβ, N, and HN secondary chemical shifts, we used a confidence parameter $\theta_f$[80] of 20. Error estimates of 0.92 ppm for Cα, 1.13 ppm for Cβ, 2.45 ppm for N, and 0.49 ppm for HN for the SPARTA+ chemical shift prediction were used in the chemical shift refinement[81]. Secondary shifts were determined using POTENCI[82]. The last fragment was not refined, and uniform weights were used. In the global reweighting step, we used a confidence parameter θ of 10. In a final step, yCTD ensembles were refined to match the measured hydrodynamic radius $R_H$ from NMR. The BioEn library[80] was used for ensemble refinement (https://github.com/bio-phys/BioEn). $R_H$ was calculated for each structure in the ensemble following the approach of Ahmed et al[83].

$10^5$ 21-mer fragments (YSPTSPS) were extracted from the yCTD ensemble. Chemical shifts and NOE contacts were computed for comparison of fragment ensembles to the NMR data of 3R-CTD. Without any refinement, we matched 29 of 36 experimental NOE contacts ≤ 5.65 Å, with two additional contacts right at cut off. With minimal further refinement (θ = 25, SKL = 0.34) 3 R structures from yCTD HCG match 33 out of 36 measured NOE contacts within the threshold and one contact just above the threshold. SK$^{Bias}$ was below 1 further indicating that importance sampling generated relevant structures. Analysis of the fit to experiment and changes in the conformer weights demonstrated good agreement with experimental chemical shifts, while staying close to the initial conformer weights. The BME2 library (https://github.com/sbottaro/BME2) was used to match the upper bound distances from NOE measurements of 3R-CTD[84,85].

Structures were analyzed using the MDAnalysis[86,87] and MDTraj Python[88] libraries.

### Molecular dynamics

Molecular dynamics (MD) simulations of full-length hCTD were carried out using the GROMACS 5.1.4 package[89,90], employing the all-atom Amber99SB-ILDN force field[77] and the TIP4P-D water model, specially optimized to study structure, dynamics and interactions of disordered proteins[91] at the atomistic level. The initial configuration of the full-length hCTD chain was generated using iTASSER[92]. Following an energy minimization, a single protein copy was simulated initially for 100 ns in a cubic water box of 20 nm x 20 nm x 20 nm in 0.15 M NaCl, from which 10 different conformations were selected at random and

placed in a cubic box of the same size with maximal possible separation between them. The effective protein concentration in the crowded multi-copy system was 2 mM (82 mg/ml), with additional NaCl ions added to a final concentration of 0.15 M (see Supplementary Table 1 for further details about the number and type of simulated molecules). Both single-copy and multi-copy simulations were then extended to a total length of 1 μs. A leap-frog algorithm was used for integration under periodic boundary conditions. In both energy minimization and production runs, neighbor-lists were updated every 10 steps, following a Verlet-scheme based grid-search approach. The bonds involving H atoms were constrained using LINCS[93]. Temperature control (T = 310 K) was achieved via a Nose-Hoover thermostat[94], with a relaxation time of 0.5 ps, while pressure (P = 1 atm) was controlled using a Parrinello-Rahman approach[95]. Compressibility for the barostat was set to 4.5 ×10−5, and the relaxation time was 10 ps. Coupling was done separately for water and protein in all cases. A twin-range spherical cut-off (1.0 nm/1.2 nm) was used for van der Waals interactions, while electrostatics were treated using the Particle-Mesh Ewald method with a real space cut-off of 1.2 nm, 0.12 nm grid, and cubic interpolation. The same simulation setup was used for LGE[28] and FUS simulations. The GROMACS simulation input files as well as the coordinates of the first and the last simulation snapshots are provided in Supplementary Data 1.

For the analysis of protein-protein interactions, the last 0.3 μs of MD trajectories were used. A distance of 3.5 Å was chosen as a cut-off for interatomic contacts, which were calculated using the *pairdist* function from the GROMACS package with a time step of 1 ns. The thus obtained all-to-all residue distance matrices for the single protein (single-copy system) or each protein-protein pair (multi-copy system) were used to derive contacts statistics (frequency per frame) and average residue interactivity along the protein sequence. This was done using scripts specially written for this purpose. Actual MD fractions for a given type of contacts were normalized by the expected fraction for this type of contacts in the randomized sequence background to get an enrichment value. For the analysis of spatial configurations of Pro-Tyr pairs, master trajectories comprising 10 ns-spaced MD snapshots for each pair with a contact frequency over the last 0.3 μs greater than 10 % were created for single- and multi-copy systems, resulting in ~15000 individual configurations in each case. Structural clustering for these master trajectories was performed using the cluster tool from the GROMACS package with applied all-atom RMSD cut-offs in the range of 0.5-1 Å. The distribution of distances between centers-of-geometry of Pro and Tyr rings (defined by heavy atoms of complete Pro residue and Tyr side-chain, respectively) were calculated for the Pro-Tyr master trajectories using *pairdist*. Protein structures were visualized using PyMol.

### Reporting summary

Further information on research design is available in the Nature Portfolio Reporting Summary linked to this article.

## Data availability

The data that support this study are available from the corresponding author upon request. CryoEM structures used in this manuscript for analysis are publicly available at the Protein Data Bank (PDB) under the code 7ENC. Source data are provided with this paper.

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

## Acknowledgements

We thank Kerstin Overkamp for HPLC purification of CTD proteins. We thank Benjamin Frühbauer for performing MD simulations of FUS. M.Z. and P.C. were supported by the Deutsche Forschungsgemeinschaft (SPP2191, project ZW 71/9-1). M.Z. was also supported by the European Research Council (ERC) under the EU Horizon 2020 research and innovation programme (grant agreement No. 787679). M.Z. and B.Z. were supported by the VolkswagenStiftung (Project-ID AZ 98188). L.S.S. thanks ReALity (Resilience, Adaptation and Longevity), M³ODEL (Mainz Institute of Multiscale Modeling) and Forschungsinitiative des Landes Rheinland-Pfalz for their support. A.C. is supported by M³ODEL. L.S.S. and A.C. gratefully acknowledge the computing time granted on the supercomputer Mogon at Johannes Gutenberg University Mainz (hpc.uni-mainz.de).

## Author contributions

D.F.S. performed protein expression and purification, recorded, processed, and analyzed NMR data, and performed phase separation assays, microscopy, FRAP, and Rosetta structure calculations. I.P.L. performed protein expression and purification, and NMR assignments of yCTD as well as CTD peptides. M.M., A.A.P., and B.Z. performed and analyzed molecular dynamics simulations. A.C., L.M.P., and L.S.S. performed and analyzed hierarchical chain growth calculations. M.B. prepared wild-type and mutant (Y1F) hCTD, and FRAP and performed phase separation experiments. J.W. prepared the human Mediator complex. P.C. supervised the preparation of hCTD variants and Mediator complex. The manuscript was prepared with input from all authors. M.Z. designed the project.

## Funding

## Competing interests

The authors declare no competing interests.
