## [Peer Review File · Nature Communications]

REVIEWER COMMENTS

Reviewer #1 (Remarks to the Author):

Flores-Solis and coworkers report that the CTD of RNA pol II is able to phase separate through interactions of the repeating tyrosine and proline residues that are characteristic of the repeating units of RNA pol II's CTD. They provide experimental data and a complimentary set of biomolecular simulations in support of these conclusions. This is a novel and important insight and would be suitable for publication in Nature communications with revisions. In particular, additional work should be performed to validate the CH-pi interaction as an important factor driving phase separation. Appropriate analytical data needs to be provided to describe the systems that they're using to claim the CH-pi interaction is important for phase separation. And the interpretation of some experiments (e.g., FRAP as a measurement of viscoelasticity) is concerning and should be addressed in these revisions. I note below major concerns that should be addressed before publication. I also note minor concerns that I think the authors should consider addressing, although I appreciate that the additional efforts might be beyond the scope of reasonable expectations for a single study.

Major Concerns:

1) The authors present data on the chemical interactions found between purified peptides and proteins, but they do not provide data on the purity or sufficient proof of their preparation despite their use of a compliment of purification techniques. It would add significantly to the rigor and validity of these experiments to include these analytical data with little additional work as they state in the methods that they recorded these data to confirm their findings. Including mass spectra and HPLC chromatograms for proteins and peptides when indicated in the methods and recording these data elsewhere when feasible. Modern mass spectrometry approaches are inherently user friendly and QTOF or MALDI data to support their biochemical experiments is appropriate given the chemical precision of techniques that their conclusions are built upon. This is especially critical for those materials labeled with ^{15}N and $^{15}\text{N}/^{13}\text{C}$ nuclides, which are analyzed with nuclear magnetic resonance spectroscopy (NMR) and form the backbone of this body of work.

2) The authors present FRAP data as a measurement of viscoelasticity (line 208-line 211). Viscoelasticity is an emergent property of condensates, but is not assessed through FRAP. A FRAP measurement assesses the diffusivity of particles in a condensate and its viscosity (through application of the Stokes-Einstein relation). To claim regularity in condensate viscoelasticity, it is necessary to measure the deformation of the condensates under force, e.g. through micro rheology.

3) As CH- π interactions have been established by research groups sophisticated in physical organic chemistry to occur between an electron rich aromatic ring and a CH moiety (see Kiessling and Diehl, ACS Chem. Biol. 2021 for a timely review). The work presented here concludes, using a variety of biophysical techniques, that CH- π interactions drive the phase separation of human and yeast CTD of RNA pol II. In this work they employ a peptide model system, which allows them to access fragments of the CTD with shorter peptides that contain the conserved heptad repeats. Depletion of electron density in the aromatic ring of tyrosine residues should result in a decrease in the CTD's ability to phase separate. Substitution of tyrosine toward phenylalanine does not appear to be sufficient despite the reduced electron density found on this ring. However, substitution of tyrosine toward a more electron density poor ring system should if the CH- π interaction is indeed driving phase separation. Using the synthetic model peptide systems described in Fig. 2, Fig. S3, and Fig S4), substitution of tyrosine to a more electron poor aromatic ring, such as with 4-fluorophenylalanine, would impact the phase separating capabilities of these model peptides, if noncovalent interactions between a particular tyrosine and other amino acids are important. Additionally, fluorine provides a sensitive NMR probe, which could be expeditious towards providing details.

Minor Concerns:

4) The authors could provide interesting insights into the nature of states sampled by the CTD of RNA Pol II during assemblage and association of the transcriptional apparatus by inclusion of the mediator complex in their assays of phase separation, fluorescence recovery, and dynamic light scattering with tyrosine mutants of the CTD. As some of these tyrosine mutants are unable to fully satisfy the CH- π interactions the authors claim important for the assemblage and function of this complex and thus they provide important controls. The authors note this (Fig. 4 and S6), however, they provide only droplet partitioning data to explore how the γ CTD tyrosine mutants interact with the mediator complex. Additional experiments that dissect the effects of the mediator complex will prove to better support their work, including an assessment of its capacity to promote condensate formation at different concentrations of mediator/CTD-mutant (providing a connection to thermodynamics) and at those concentrations where phase separation is accessible and assessment of diffusivity (through FRAP).

5) How the oligomeric particles identified in solutions of tyrosine γ CTD mutants are interacting with the mediator complex and to what degree could provide insights into the early stages of transcriptional apparatus assembly (before all or some critical number of interactions is satisfied) (Fig. 4, Fig. S6). Dynamic light scattering provides an indication of different particle sizes in solution and using thoughtful controls, whether these oligomeric particles are complexing with mediator or not should be observable by asking if the 'oligomeric particles' are still present or are lost upon mixing with the mediator complex. These data could provide interesting insights into the nature of states sampled by the CTD of RNA pol II during assemblage of the transcriptional apparatus and inclusion of mediator complex, because these 'oligomeric' particles cannot fully satisfy the number of contacts needed to induce phase separate.

6) CH- π interactions have been established to involve dispersion interactions between the CH group and the π -density of an electron enriched aromatic ring. It has been established in the physical organic chemistry literature (See work from the Wennemers, Raines, and Zondlo labs), modified prolines suitable for solid phase peptide synthesis are commercially accessible and can be employed to synthesize model peptides with desirable physical properties. If CH- π interactions are indeed important for phase separation of these systems, then the authors should be able to demonstrate through use of 4-(S) and 4-(R) modified proline residues (perhaps, with fluorine for more direct application towards NMR experiments) that the energetics of these interactions can be lost due to changes in the ability of the proline to engage in CH- π interactions. A caveat here is that these effects influence the n-to- π^* interaction, and therefore the cis/trans ratio of proline residues as observed in isolation.

Reviewer #2 (Remarks to the Author):

The manuscript by Flores-Solis et al. proposes a series of molecular interactions that drive the phase separation of the carboxy-terminal domain (CTD) of RNA polymerase (pol) II. The CTD of RNA Pol II contains 52 heptad repeats of the consensus sequence YSPTSPS with a divergent C-terminal region. The authors demonstrate that the CTD of RNA Pol II exhibit LCST-phase behaviour, with increasing phase separation propensity at higher temperatures, salt-concentrations and at pHs closer to the theoretical pI of the IDR. ^1H - ^1H NOE NMR measurements of the human CTD in the dilute phase identify intramolecular contacts between tyrosine ring protons and the side-chain protons of serines, prolines and threonines. Notably, these contacts appear to be preserved in NMR samples containing an inhomogeneous mixture of dilute and condensed phases of CTD, ruling in the possibility that such tyrosine – serine/threonine/proline side-chain interactions are important for phase separation. In support, mutations of tyrosines to leucines or serines completely inhibited phase separation, whereas phenylalanine mutations largely preserved phase separation capacity.

The authors demonstrate that residues within the perfect heptad repeats see nearly identical chemical environments leading to overlapping resonances within NMR spectra, and are aptly mimicked by model peptides containing 1-6 repeats (1R-CTD – 6R-CTD) of consensus sequence. Thus, backbone chemical shift, ^1H - ^1H NOE and residual dipolar coupling measurements on the 3R-CTD were used as restraints for CS-Rosetta based structural calculations and the ensemble generated are consistent with a highly dynamic, largely extended conformation with some propensity for turn conformations. Notably, based on such structures and independent all-atom MD simulations, the authors highlight the prevalence of intermolecular tyrosine interactions with other tyrosines, serines, threonines and prolines, with a marginal bias for prolines over serines and threonines and significantly higher prevalence of tyrosine self-interactions. In addition, the authors show co-phase separation of RNA Pol II CTD with the human Mediator complex that is strongly dependent on the tyrosine content of CTD, further pointing to the importance of tyrosines in phase separation.

Overall, using a combination of microscopy, NMR, DLS and all-atom MD simulations of various CTD and model peptide constructs, the authors highlight the importance of multivalent tyrosine interactions for driving the LCST phase separation of RNA Pol II CTD and its co-phase separation with the human Mediator Complex. Such experimental insights add to a sizeable body of evidence implying a crucial role of the CH/ π character of tyrosines and other aromatic residues for promoting phase separation. The specific importance of tyrosine-proline interactions, which the authors suggest are “enriched” in condensed states of CTD, is not adequately supported by the data in the manuscript, as described in greater detail below. The work detailed in this manuscript, while advancing our understanding of the driving forces behind RNA Pol II CTD phase separation, requires additional experiments before it can appeal to the broad readership of Nature Communications.

Major Concerns/comments:

1. The authors imply that tyrosine-proline interactions drive the temperature- and concentration-dependent phase separation of RNA polymerase II CTD (line 25) and are enriched in condensed states (line 27). However, these claims are not adequately supported by experimental evidence, as described below. ¹H-¹H NOESY experiments in the dilute phase indicate the presence of intramolecular side-chain contacts between tyrosines and all other amino acid types in the CTD (Fig. 1d, 2c). The tyrosine-proline intramolecular NOEs shown in Fig. 2c do not appear to be significantly more intense compared to other amino acids. Moreover, it is unclear whether NOEs between other amino acid pairs are also observed for the CTD, e.g. Pro-Ser, Pro-Thr, Ser-Thr etc., as the authors neither show the spectra nor mention the absence/presence of their existence. In the case of the model peptide (3R-CTD), the authors indeed claim that Ser-Thr and Pro-Ser contacts are observed, albeit again no spectra are shown. Thus, the unique role of tyrosine-proline interactions is not sufficiently highlighted.

Moreover, the authors do not provide experimental insight into intermolecular contacts between CTD protomers, which arguably are critical, and likely more pertinent, in driving phase separation. While the ¹H-¹H NOEs observed in the “phase-separated” state (a mixture of protein in dilute and condensed phases) may arise from both intra- and inter-molecular contacts, they do not “demonstrate the enrichment of tyrosine-proline interactions in condensed states”, as like the dilute phase data there is an abundance of tyrosine contacts with all other amino acid types.

While the authors highlight the importance of multivalent tyrosine (aromatic) interactions driving phase separation of CTD, as evidenced by the Y1L and various patterns of Y1S mutations inhibiting phase separation but not Y1F (Fig. 4f), the importance of tyrosines for phase separation has already been extensively demonstrated (PMID: 29961577, PMID: 28924037, PMID: 32029630, PMID: 32312191). Instead, mutations of individual or combination of proline residues and assessment of the impact of such mutations on phase behavior would have been more valuable.

Furthermore, the 1 μ s MD simulations performed in a crowded context (10 copies of CTD) show only a marginal preference for P-Y interaction pairs (14%) in the pool of all contacts detected (P-S = 14%, S-Y = 12% etc.). As well, the interaction patterns for heptad positions identify Y1-P6 contacts (1.9) and Y1-P3

(1.7) to be marginally higher than other Y-X contacts (ranging from 1.3-1.7), with Y-Y contacts being the most common (2.7).

Certainly, tyrosine-proline interactions may be one interaction type involved in the phase separation of CTD and other LCST phase-separating proteins, but its significance does not appear as great as the authors explicitly claim in the manuscript. Rather, what appears to be consistent with the literature is the importance of interactions between aromatic amino acids and non-polar groups of both polar (serine, threonine) and non-polar (proline) amino acids (PMID: 31270472 and PMID: 32312191).

2. The authors used experimental chemical shifts, RDCs and NOEs on model peptide 3R-CTD to generate structures using Rosetta and hierarchical chain growth calculations. The ensembles generated from such experimental restraints exhibit short O-N distances between S-2P-1S-0Y1 which the authors suggest are indicative of the presence of turn conformations that bring together tyrosine and proline rings (line 198-201, Fig. 3b,c). The methods section regarding the Rosetta structure calculations is scarce, and it is unclear how many structures remained after filtering using Rosetta all-atom energy functions. Is it the 13 shown in Fig. 3b,c? Moreover, it is ambiguous what the authors are implying in line 200, where they state “indicative of turns populated by 15.8%”. How was this value calculated?

The authors show good agreement between hydrodynamic radii calculated from NMR, DLS and the structural ensemble generated from Rosetta (Fig.3a), which presumably the authors use as a benchmark to determine if the structures faithfully recapitulate the experimental restraints. It is unclear whether the hydrodynamic radii alone are sufficient to gauge how representative the structures are of conformers in solution and whether these turns are formed. A more rigorous attempt to validate the presence of such turn conformations is necessary. Especially valuable would be the measurement of backbone torsion angles and NH-NH NOEs, which should be feasible for the model 3R-CTD.

3. The DLS experiments in Fig. 1c indicate the formation of larger particles with increasing temperature and salt concentrations. However, without accompanying microscopy images it is difficult to know the morphology of these larger particles. Are they percolated clusters, aggregates, or liquid-like droplets?

4. The authors should provide photos of the uniformly mixed and “phase-separated” NMR samples as opposed to a cartoon representation currently shown in Fig. 1e.

Minor Concerns/comments:

1. The streak of interresidue correlations observed in 1H-1H NOESY spectra of the “phase-separated” state may also arise from differences in the magnetic susceptibility of the various droplets formed at elevated temperatures compared to the uniformly mixed sample.

2. The y-axis label of Fig. S1c is incorrectly spelled.

Reviewer #3 (Remarks to the Author):

Flores-Solis et al. study the phase separation of RNA Pol II CTD using a combination of NMR spectroscopy and a variety of computational techniques (ensemble enumeration and MD simulations). The central goal is to identify the molecular features responsible for its phase separation and the co-recruitment of the human Mediator complex. The central conclusion of the work is that tyrosine-proline interactions are broadly important for IDP phase separation.

Overall, this work should be of interest to people interested in learning more about the rules of protein phase separation and the relative contribution of different interaction pairs or modes. I have following comments and suggestions for the authors.

(a) The authors emphasize at many places that "tyrosine-proline interactions being the driver and their abundance in other LC protein condensed states." These statements should be better qualified as the experimental data in the paper is not directly evaluating the relative contributions of different amino acid pairings for RNA Pol II CTD or other LC proteins. Furthermore, NMR data clearly indicates the role of other pairs involving polar residues which is consistent with emerging view on molecular interactions present in the condensed phase.

(b) Please recheck the citations associated with the following sentence: "The LCST behaviour of CTD is in agreement with a low content of charged amino acids." I would have expected papers discussing temperature-controlled phase separation of proteins including the role of sequence composition in UCST vs. LCST behavior.

(c) NMR figures can be labeled better to avoid having to read the main text to understand the peaks, etc.

(d) Some statements such as "Some of the strongest signals were seen between..." require quantification. I am not an expert so cannot judge how the specific peaks compare with each other. Please check the paper carefully for such statements and add appropriate numbers for comparison.

(e) The authors should provide a bit more discussion on the counterintuitive result obtained for Y1F variant in the context of current literature on phase separation. Previously, many similar mutants for other proteins such as FUS LC, LAF-1 RGG resulted in no phase separation at conditions where Tyr-variants formed droplets. Similarly, the authors start to discuss the correspondence between single-chain expansion/collapse and phase separation in the context of new variants with different distribution of Tyr residues, but do not connect it with the existing literature.

(f) I was a bit surprised that the authors use a slightly non-standard multi-copy simulation setup where chains may not be properly relaxed. Can the authors comment on why and how the results may differ from the commonly used slab geometry?

Reviewer #1:

Flores-Solis and coworkers report that the CTD of RNA pol II is able to phase separate through interactions of the repeating tyrosine and proline residues that are characteristic of the repeating units of RNA pol II's CTD. They provide experimental data and a complimentary set of biomolecular simulations in support of these conclusions. This is a novel and important insight and would be suitable for publication in Nature communications with revisions. In particular, additional work should be performed to validate the CH- π interaction as an important factor driving phase separation. Appropriate analytical data needs to be provided to describe the systems that they're using to claim the CH- π interaction is important for phase separation. And the interpretation of some experiments (e.g., FRAP as a measurement of viscoelasticity) is concerning and should be addressed in these revisions. I note below major concerns that should be addressed before publication. I also note minor concerns that I think the authors should consider addressing, although I appreciate that the additional efforts might be beyond the scope of reasonable expectations for a single study.

Reply: We thank the reviewer for the positive assessment of our work and for the helpful suggestions for improvements.

The authors present data on the chemical interactions found between purified peptides and proteins, but they do not provide data on the purity or sufficient proof of their preparation despite their use of a compliment of purification techniques. It would add significantly to the rigor and validity of these experiments to include these analytical data with little additional work as they state in the methods that they recorded these data to confirm their findings. Including mass spectra and HPLC chromatograms for proteins and peptides when indicated in the methods and recording these data elsewhere when feasible. Modern mass spectrometry approaches are inherently user friendly and QTOF or MALDI data to support their biochemical experiments is appropriate given the chemical precision of techniques that their conclusions are built upon. This is especially critical for those materials labeled with ^{15}N and ^{13}C , which are analyzed with NMR and form the backbone of this body of work.

Reply: We thank the referee for the constructive feedback. In the revised version of the manuscript, we added the corresponding chromatograms and mass spectrometry data for each protein and peptide (Supplementary Fig. 1). For the peptide data, chromatograms and mass spectra were adapted from the data sent by our peptide supplier (GenScript).

The authors present FRAP data as a measurement of viscoelasticity (line 208-line 211). Viscoelasticity is an emergent property of condensates, but is not assessed through FRAP. A FRAP measurement assesses the diffusivity of particles in a condensate and its viscosity (through application of the Stokes-Einstein relation). To claim regularity in condensate viscoelasticity, it is necessary to measure the deformation of the condensates under force, e.g. through microrheology.

Reply: We agree that a proper term for our observations is the apparent diffusivity rather than the implication of the viscoelasticity. We adapted the text avoiding the term not to cause misinterpretation of our data. Measuring the viscoelastic properties of the droplets would add valuable information to fully characterize the different *in vitro* systems in our manuscript. Unfortunately, we are currently not able to run those experiments.

CH- π interactions have been established by research groups sophisticated in physical organic chemistry to occur between an electron rich aromatic ring and a CH moiety (see Kiessling and Diehl, ACS Chem. Biol. 2021 for a timely review). The work presented here concludes, using a variety of biophysical techniques, that CH- π interactions drive the phase separation of human and yeast CTD of RNA pol II. In this work they employ a peptide model system, which allows them to access fragments of the CTD with shorter peptides that contain the conserved heptad repeats. Depletion of electron density in the aromatic ring of tyrosine residues should result in a decrease in the CTD's ability to phase separate. Substitution of tyrosine toward phenylalanine does not appear to be sufficient despite the reduced electron density found on this ring. However, substitution of tyrosine toward a more electron density poor ring system should if the CH- π interaction is indeed driving phase separation. Using the synthetic model peptide systems described in Fig. 2, Fig. S3, and Fig S4), substitution of tyrosine to a more electron poor aromatic ring, such as with 4-fluorophenylalanine, would impact the phase separating capabilities of these model peptides, if noncovalent interactions between a particular tyrosine and other amino acids are important. Additionally, fluorine provides a sensitive NMR probe, which could be expeditious towards providing details.

Reply: We thank the reviewer for the thoughtful suggestions, and agree that subtracting electron density from the rings of tyrosine will further demonstrate the aromatic character of the phase separation and set the threshold for this interaction from the Tyr perspective. We also agree that fluorine is an excellent NMR probe in terms of sensitivity and dispersion of the peaks. Because of the already broad nature of the study including peptide studies, complementary experiments with both hCTD and yCTD, and co-recruitment experiments with Mediator complex, we believe that some of the suggested approaches are beyond the scope of the current work.

However, to gain further insights into the contribution of the specific sequence of the CTD to its ability to phase separate, we prepared two designed yCTD mutants where the Pro residues are swapped (TPPS and PYP) when compared to the wild-type sequence. The new yCTD mutants were subjected to phase separation and NMR experiments. The new data are shown in Fig. 4h,I and S7, and discussed in the revised version of the manuscript.

The authors could provide interesting insights into the nature of states sampled by the CTD of RNA Pol II during assemblage and association of the transcriptional apparatus by inclusion of the mediator complex in their assays of phase separation, fluorescence recovery, and dynamic light scattering with tyrosine mutants of the CTD. As some of these tyrosine mutants are unable to fully satisfy the CH- π interactions the authors claim important for the assemblage and function of this complex and thus they provide important controls. The authors note this (Fig. 4 and S6), however, they provide only droplet partitioning data to explore how the yCTD tyrosine mutants interact with the mediator complex. Additional experiments that dissect the effects of the mediator complex will prove to better support their work, including an assessment of its capacity to promote condensate formation at different concentrations of mediator/CTD-mutant (providing a connection to thermodynamics) and at those concentrations where phase separation is accessible and assessment of diffusivity (through FRAP).

Reply: We thank the reviewer for the suggestions. For the original submission of our manuscript, we evaluated the co-recruitment of hMED and CTD mutants in different conditions. However, in the particular case of the Y1S mutants, we could not observe the recruitment of any Y1S mutant at any given concentration into hMED droplet, i.e. the suggested FRAP experiments for the CTD-mutants are not possible. Instead, we recorded FRAP data for the interaction between hCTD and hMED at different hCTD:hMED molar ratios. In these experiments, we observed a decreased recovery rate of the bleached fluorescence signal of hCTD at equimolar hMED concentration (Supplementary Fig. 9f).

How the oligomeric particles identified in solutions of tyrosine yCTD mutants are interacting with the mediator complex and to what degree could provide insights into the early stages of transcriptional apparatus assembly (before all or some critical number of interactions is satisfied) (Fig. 4, Fig. S6). Dynamic light scattering provides an indication of different particle sizes in solution and using thoughtful controls, whether these oligomeric particles are complexing with mediator or not should be

observable by asking if the ‘oligomeric particles’ are still present or are lost upon mixing with the mediator complex. These data could provide interesting insights into the nature of states sampled by the CTD of RNA pol II during assemblage of the transcriptional apparatus and inclusion of mediator complex, because these ‘oligomeric’ particles cannot fully satisfy the number of contacts needed to induce phase separate.

Reply: This is a stimulating idea that we tried experimentally. However, we conducted those experiments without success since the oligomeric particles of the Y1S mutants tend to be broadly dispersed. Additionally, we could not solve some of the technical problems to filter the intense signal coming from the hMED complex and the oligomeric Y1S mutants.

CH- π interactions have been established to involve dispersion interactions between the CH group and the π -density of an electron enriched aromatic ring. It has been established in the physical organic chemistry literature (See work from the Wennemers, Raines, and Zondlo labs), modified prolines suitable for solid phase peptide synthesis are commercially accessible and can be employed to synthesize model peptides with desirable physical properties. If CH- π interactions are indeed important for phase separation of these systems, then the authors should be able to demonstrate through use of 4(S) and 4-(R) modified proline residues (perhaps, with fluorine for more direct application towards NMR experiments) that the energetics of these interactions can be lost due to changes in the ability of the proline to engage in CH- π interactions. A caveat here is that these effects influence the n-to- π^* interaction, and therefore the cis/trans ratio of proline residues as observed in isolation.

Reply: We thank the reviewer for the suggestion. Indeed, these experiments may address deeper aspects of CH- π interactions in the context of intramolecular interactions in CTD peptides. However, none of the CTD peptides phase separated even at the highest concentration tested, such that we would not be able to connect changes in intramolecular interactions with the ability to phase separate. Additionally, introduction of modified proline residues into yCTD/hCTD is highly challenging. We therefore opted for the rationalized scrambling of the sequence of yCTD and performed both phase separation and NMR experiments (please see above).

Reviewer #2:

The manuscript by Flores-Solis et al. proposes a series of molecular interactions that drive the phase separation of the carboxy-terminal domain (CTD) of RNA polymerase (pol) II. The CTD of RNA Pol II contains 52 heptad repeats of the consensus sequence YSPTSPS with a divergent C-terminal region. The authors demonstrate that the CTD of RNA Pol II exhibit LCST-phase behavior, with increasing phase separation propensity at higher temperatures, salt-concentrations and at pHs closer to the theoretical pI of the IDR. ^1H - ^1H NOE NMR measurements of the human CTD in the dilute phase identify intramolecular contacts between tyrosine ring protons and the side-chain protons of serines, prolines and threonines. Notably, these contacts appear to be preserved in NMR samples containing an inhomogeneous mixture of dilute and condensed phases of CTD, ruling in the possibility that such tyrosine – serine/threonine/proline side-chain interactions are important for phase separation. In support, mutations of tyrosines to leucines or serines completely inhibited phase separation, whereas phenylalanine mutations largely preserved phase separation capacity. The authors demonstrate that residues within the perfect heptad repeats see nearly identical chemical environments leading to overlapping resonances within NMR spectra, and are aptly mimicked by model peptides containing 16 repeats (1R-CTD – 6R-CTD) of consensus sequence. Thus, backbone chemical shift, 1H-1H NOE and residual dipolar coupling measurements on the 3R-CTD were used as restraints for CS-Rosetta

based structural calculations and the ensemble generated are consistent with a highly dynamic, largely extended conformation with some propensity for turn conformations. Notably, based on such structures and independent all-atom MD simulations, the authors highlight the prevalence of intermolecular tyrosine interactions with other tyrosines, serines, threonines and prolines, with a marginal bias for prolines over serines and threonines and significantly higher prevalence of tyrosine self-interactions. In addition, the authors show co-phase separation of RNA Pol II CTD with the human Mediator complex that is strongly dependent on the tyrosine content of CTD, further pointing to the importance of tyrosines in phase separation. Overall, using a combination of microscopy, NMR, DLS and all-atom MD simulations of various CTD and model peptide constructs, the authors highlight the importance of multivalent tyrosine interactions for driving the LCST phase separation of RNA Pol II CTD and its co-phase separation with the human Mediator Complex. Such experimental insights add to a sizeable body of evidence implying a crucial role of the CH/ π character of tyrosines and other aromatic residues for promoting phase separation. The specific importance of tyrosine-proline interactions, which the authors suggest are “enriched” in condensed states of CTD, is not adequately supported by the data in the manuscript, as described in greater detail below. The work detailed in this manuscript, while advancing our understanding of the driving forces behind RNA Pol II CTD phase separation, requires additional experiments before it can appeal to the broad readership of Nature Communications.

Reply: We thank the reviewer for the careful evaluation of our manuscript and for the helpful suggestions for improvements.

The authors imply that tyrosine-proline interactions drive the temperature- and concentration-dependent phase separation of RNA polymerase II CTD (line 25) and are enriched in condensed states (line 27). However, these claims are not adequately supported by experimental evidence, as described below. ^1H - ^13C NOESY experiments in the dilute phase indicate the presence of intramolecular side-chain contacts between tyrosines and all other amino acid types in the CTD (Fig. 1d, 2c). The tyrosine-proline intramolecular NOEs shown in Fig. 2c do not appear to be significantly more intense compared to other amino acids. Moreover, it is unclear whether NOEs between other amino acid pairs are also observed for the CTD, e.g. Pro-Ser, Pro-Thr, Ser-Thr etc., as the authors neither show the spectra nor mention the absence/presence of their existence. In the case of the model peptide (3R-CTD), the authors indeed claim that Ser-Thr and Pro-Ser contacts are observed, albeit again no spectra are shown. Thus, the unique role of tyrosine-proline interactions is not sufficiently highlighted. Moreover, the authors do not provide experimental insight into intermolecular contacts between CTD protomers, which arguably are critical, and likely more pertinent, in driving phase separation. While the ^1H - ^1H NOEs observed in the “phase-separated” state (a mixture of protein in dilute and condensed phases) may arise from both intra- and inter-molecular contacts, they do not “demonstrate the enrichment of tyrosine-proline interactions in condensed states”, as like the dilute phase data there is an abundance of tyrosine contacts with all other amino acid types. While the authors highlight the importance of multivalent tyrosine (aromatic) interactions driving phase separation of CTD, as evidenced by the Y1L and various patterns of Y1S mutations inhibiting phase separation but not Y1F (Fig. 4f), the importance of tyrosines for phase separation has already been extensively demonstrated (PMID: 29961577, PMID: 28924037, PMID: 32029630, PMID: 32312191). Instead, mutations of individual or combination of proline residues and assessment of the impact of such mutations on phase behavior would have been more valuable. Furthermore, the 1 μs MD simulations performed in a crowded context (10 copies of CTD) show only a marginal preference for P-Y interaction pairs (14%) in the pool of all contacts detected (P-S = 14%, S-Y = 12% etc.). As well, the interaction patterns for heptad positions identify Y1-P6 contacts (1.9) and Y1-P3 (1.7) to be marginally higher than other Y-X contacts (ranging from 1.3-1.7), with Y-Y contacts being the most common (2.7).

Reply: We thank the reviewer for these comments. Please note that the raw contact statistics are strongly affected by sequence bias: the more residues of a given type there are in a sequence, the more they will dominate the contact statistics on average. To remove this bias, it is best to normalize the observed contact fractions by what is expected at random, based on the abundances of the residues in question, and compare the corresponding enrichment values. Based on this analysis, Pro-Tyr contacts are altogether the second most enriched type of contacts (1.8 times above background) after Tyr-Tyr (2.7x) in the multicopy system, while Pro-Ser contacts are depleted (0.7 of the background level) (Supplementary Fig. 8b).

Certainly, tyrosine-proline interactions may be one interaction type involved in the phase separation of CTD and other LCST phase-separating proteins, but its significance does not appear as great as the authors explicitly claim in the manuscript. Rather, what appears to be consistent with the literature is the importance of interactions between aromatic amino acids and non-polar groups of both polar (serine, threonine) and non-polar (proline) amino acids (PMID: 31270472 and PMID: 32312191).

Reply: We thank the reviewer for these thoughtful comments. We also agree with the reviewer that our data are better described taking into account more broadly the spectrum of interactions, and not associating terms such as “driving phase separation” to only Tyr-Pro interactions. Indeed, both the NMR and MD simulation data reveal the presence of a wide range of interactions, including Tyr-Pro contacts. NMR spectra of 3R-CTD peptide and γ CTD, displaying intramolecular NOE contacts to tyrosine, are shown in Figs. 2c, 4h and S4a,b. Regarding the experimental detection of intermolecular NOEs, this is quite challenging. We accordingly modified the different sections of the manuscript, including title, abstract and discussion.

To provide further analysis of the contribution of the CTD amino acid sequence to the protein's ability to phase separate, we prepared two designed CTD variants (YPSTSSP named PYP, and YSTPPSS named TPPS; Fig. 4g). The two variants have the same amino acid composition, but with different proximity of proline to tyrosine. For TPPS, prolines in the canonical heptad in positions 3 and 6 are swapped with residues in positions 4 and 5 (Thr and Ser), respectively, producing the new heptad YSTPPSS. Similarly, the heptad of the PYP variant interchanges the proline residues in positions 3 and 5 with positions 2 and 7 (Ser) resulting in the heptad YPSTSSP. DLS and fluorescence microscopy showed that the two CTD variants phase separate at 150 mM NaCl with increasing temperature in contrast to wild-type CTD (Supplementary Fig. 7). In addition, they form more and/or larger droplets at 500 mM NaCl, in particular at 25 °C. At 1000 mM NaCl, i.e. at very high ionic strength, both variants phase separate at 15 °C in contrast to wild-type γ CTD. Additionally, larger droplets were observed by fluorescence microscopy at room temperature for the PYP variant (Supplementary Fig. 7). Probing the diffusivity of droplets formed by the three proteins using fluorescence recovery after photobleaching (FRAP) showed similar fluorescence recovery rates for γ CTD and the TPPS variant, while the PYP variant displayed decreased diffusivity (Supplementary Fig. 7d). The data demonstrate that the specific sequence of amino acids in the heptad repeat influences CTD's ability to phase separate into liquid-like droplets, and affect their molecular properties.

We then analyzed by NMR intramolecular contacts in the dilute state of the two variants and compared them to wild-type CTD. To this end, we recorded two-dimensional ^1H - ^1H NOESY spectra and analyzed the signal intensities of the cross peaks involving aromatic tyrosine protons (Fig. 4h-j). For both CTD variants, lower cross peak intensities were present for the epsilon position of the tyrosine ring (Fig. 4i,j). Additionally, the TPPS variant showed lower cross peak intensities for the delta position (Fig. 4j). The combined observation of enhanced phase separation and reduced intramolecular tyrosine contacts, including Tyr-Pro contacts, of the two sequence-perturbed variants suggests that intramolecular contacts involving the aromatic ring of tyrosine compete with intermolecular contacts driving CTD phase separation.

The authors used experimental chemical shifts, RDCs and NOEs on model peptide 3R-CTD to generate structures using Rosetta and hierarchical chain growth calculations. The ensembles generated from such experimental restraints exhibit short O-N distances between S-2P-1S-0Y1 which the authors suggest are indicative of the presence of turn conformations that bring together tyrosine and proline rings (line 198-201, Fig. 3b,c). The methods section regarding the Rosetta structure calculations is scarce, and it is unclear how many structures remained after filtering using Rosetta all-atom energy functions. Is it the 13 shown in Fig. 3b,c? Moreover, it is ambiguous what the authors are implying in line 200, where they state “indicative of turns populated by 15.8%”. How was this value calculated?

Reply: Thanks for the suggestions/questions. In the revised version of the manuscript, we provided further details regarding the Rosetta structure calculation. We also rephrased the sentence stating the population percentage of turns.

The authors show good agreement between hydrodynamic radii calculated from NMR, DLS and the structural ensemble generated from Rosetta (Fig.3a), which presumably the authors use as a benchmark to determine if the structures faithfully recapitulate the experimental restraints. It is unclear whether the hydrodynamic radii alone are sufficient to gauge how representative the structures are of conformers in solution and whether these turns are formed. A more rigorous attempt to validate the presence of such turn conformations is necessary. Especially valuable would be the measurement of backbone torsion angles and NH-NH NOEs, which should be feasible for the model 3R-CTD.

Reply: We thank the reviewer for the suggestion. To implement the suggestion, we added graphs to Fig. S4e that show different NOEs (including HN-HN NOEs) in 3R-CTD. These cross-peaks are in agreement with the presence of β -turns (according to the work published by M.M. Harding, DOI: <https://doi.org/10.1021/jm00103a002>). Signal overlap limits the number of NOEs that could be analyzed. Please also note that the small magnitude of the RDC values of Y8 and Y15 (Fig. 2d) are indicative of turn formation. Additionally, dihedral angle restraints derived from chemical shifts were used in the Rosetta structure calculation, ensuring that the resulting structures satisfy them.

The DLS experiments in Fig. 1c indicate the formation of larger particles with increasing temperature and salt concentrations. However, without accompanying microscopy images it is difficult to know the morphology of these larger particles. Are they percolated clusters, aggregates, or liquid-like droplets?

Reply: Thanks for the suggestion. We have added microscopy images as Fig. 1d. The images indicate that liquid-like droplets are present.

The authors should provide photos of the uniformly mixed and “phase-separated” NMR samples as opposed to a cartoon representation currently shown in Fig. 1e.

Reply: As suggested, we included the photos in Fig. 1f.

The streak of interresidue correlations observed in ^1H - ^1H NOESY spectra of the “phase-separated” state may also arise from differences in the magnetic susceptibility of the various droplets formed at elevated temperatures compared to the uniformly mixed sample.

Reply: We agree, and modified the statement to: “*The observed heterogeneity in chemical shifts might be due to a combination of inhomogeneities inside the condensate and associated*”

exchange processes plus different magnetic susceptibilities from emerging droplets in the sample.”

The y-axis label of Fig. S1c is incorrectly spelled.

Reply: Thanks for spotting. We corrected it.

Reviewer #3:

Flores-Solis et al. study the phase separation of RNA Pol II CTD using a combination of NMR spectroscopy and a variety of computational techniques (ensemble enumeration and MD simulations). The central goal is to identify the molecular features responsible for its phase separation and the co-recruitment of the human Mediator complex. The central conclusion of the work is that tyrosine-proline interactions are broadly important for IDP phase separation. Overall, this work should be of interest to people interested in learning more about the rules of protein phase separation and the relative contribution of different interaction pairs or modes. I have following comments and suggestions for the authors.

Reply: We thank the reviewer for the careful evaluation of our manuscript and for the helpful suggestions for improvements.

The authors emphasize at many places that "tyrosine-proline interactions being the driver and their abundance in other LC protein condensed states." These statements should be better qualified as the experimental data in the paper is not directly evaluating the relative contributions of different amino acid pairings for RNA Pol II CTD or other LC proteins. Furthermore, NMR data clearly indicates the role of other pairs involving polar residues which is consistent with emerging view on molecular interactions present in the condensed phase.

Reply: Thanks for the suggestion. As described in our reply to reviewer #2, we qualified such statements throughout the manuscript. Accordingly, we also changed the title, abstract and made some changes to the discussion.

Please recheck the citations associated with the following sentence: "The LCST behavior of CTD is in agreement with a low content of charged amino acids." I would have expected papers discussing temperature-controlled phase separation of proteins including the role of sequence composition in UCST vs. LCST behavior.

Reply: Thanks for the suggestion. We corrected it.

NMR figures can be labeled better to avoid having to read the main text to understand the peaks, etc.

Reply: We added labels to Fig. 2a. Additionally, we included the region of interest in the new panels of Fig. 4h.

Some statements such as "Some of the strongest signals were seen between..." require quantification. I am not an expert so cannot judge how the specific peaks compare with each other. Please check the paper carefully for such statements and add appropriate numbers for comparison.

Reply: Thanks for the suggestion. We again carefully checked the manuscript.

The authors should provide a bit more discussion on the counterintuitive result obtained for Y1F variant in the context of current literature on phase separation. Previously, many similar mutants for other proteins such as FUS LC, LAF-1 RGG resulted in no phase separation at conditions where Tyr-variants formed droplets. Similarly, the authors start to discuss the correspondence between single-chain expansion/collapse and phase separation in the context of new variants with different distribution of Tyr residues, but do not connect it with the existing literature.

Reply: We thank the reviewer for this suggestion. In the revised version of the manuscript, we now state: *“Notably, substitution of tyrosine for phenylalanine in the low complexity domains of the proteins Fused in Sarcoma and LAF-1 attenuates LLPS43. In contrast, replacement of tyrosine by phenylalanine in hCTD had little influence on the protein’s ability to form droplets (Fig. 4a,b). The data suggest that in case of hCTD phase separation, CH-pi and pi-pi interactions maybe more important than hydrogen bond formation for intermolecular association.”*

I was a bit surprised that the authors use a slightly non-standard multi-copy simulation setup where chains may not be properly relaxed. Can the authors comment on why and how the results may differ from the commonly used slab geometry?

Reply: Thanks for this question. The aim of the MD simulations was to study the organization of the crowded, protein-rich phase and not to model the process of its formation or probe its interface with the protein-poor phase. Indeed, many studies of LLPS phenomena by MD simulations utilize the slab configuration in which the protein-rich phase is sandwiched between two protein-poor (solvent-rich) regions. The downside of such an approach, however, is that it primarily focuses on and accurately captures the interface region between protein-rich and protein-poor phases, while providing a less accurate picture of the interior of the protein-rich phase, especially if the simulated protein-rich slab is relatively small. This is particularly relevant in simulations of large proteins, such as the Pol II CTD, whereby the sheer size of the system precludes us from simulating an extensively large, protein-rich slab. This, together with the fact that our objective is understanding the properties of the protein-rich phase, underscores our choice of the simulation setup.

Regarding relaxation, both slab configuration and protein-rich configuration face equal challenges i.e. conformational relaxation times of the simulated systems do not strongly depend on the simulation setup. The main result of our MD simulations concerns the obtained residue contact statistics in the crowded phase, for which we use the part of the trajectory (the last 0.3 μ s), where the number of partners per protein molecule (valency) reaches a stable plateau (~ 6 , see figure below), indicating that local relaxation has been reached.

Fig. Valency evolution during the “multi-copy” MD simulation.

REVIEWERS' COMMENTS

Reviewer #1 (Remarks to the Author):

The authors have addressed my concerns with their revisions and explanations. I believe this manuscript is suitable for publication.

Reviewer #2 (Remarks to the Author):

The authors have aptly addressed the major and minor comments/concerns raised and the manuscript can be accepted in its current state.

Rephrasing the original narrative to include not only Tyr-Pro interactions but also other Tyr-X interactions as drivers of phase separation of CTD of Pol II addresses the main concern. This is now further supported by NOE contact statistics for all residue pairs/atoms, providing experimental evidence that a collection of Tyrosine - X residue interactions drive CTD phase separation, including Tyr-Pro. Although the authors still do not show spectra highlighting the NOEs between residue pairs besides Tyr-X, as asked, the contact statistics provided suffice.

While intermolecular contacts were not mapped, as suggested to the authors, it is understandable given the inherent challenges with doing such measurements.

Reviewer #3 (Remarks to the Author):

The authors have satisfactorily addressed my previous comments.